# ON PAC-BAYES BOUNDS FOR DEEP NEURAL NETWORKS USING THE LOSS CURVATURE

## ABSTRACT

We investigate PAC-Bayes bounds for deep neural networks through the closely related IB-Lagrangian objective. We first propose to approximate the IB-Lagrangian through a second order Taylor expansion of the randomized loss around the minimum. For the case of Gaussian priors and posteriors with fixed means and diagonal covariance, we are able to derive a lower bound to this approximation that corresponds to an "invalid" PAC-Bayes prior (a prior that is training set dependent). Our lower bound depends only on the flatness of the minimum, and the distance between the prior and posterior means. Through a number of experiments our lower bound implies easy and and hard cases where one can or cannot prove generalization even through "cheating". Motivated by this result, we see that for the easy cases, a simple baseline closely matches our lower bound and is sufficient to achieve a non-vacuous PAC-Bayes bound. Crucially the baseline prior is centered on the random deep neural network initialization. This suggests that a good prior mean choice is the main innovation in recent non-vacuous bounds with further optimization of the PAC-Bayes bound through SGD having a complementary role.

## 1 INTRODUCTION

Recently two works Dziugaite & Roy (2017) Zhou et al. (2018) have made significant progress in proving generalization for deep neural networks. They manage to prove non-vacuous generalization

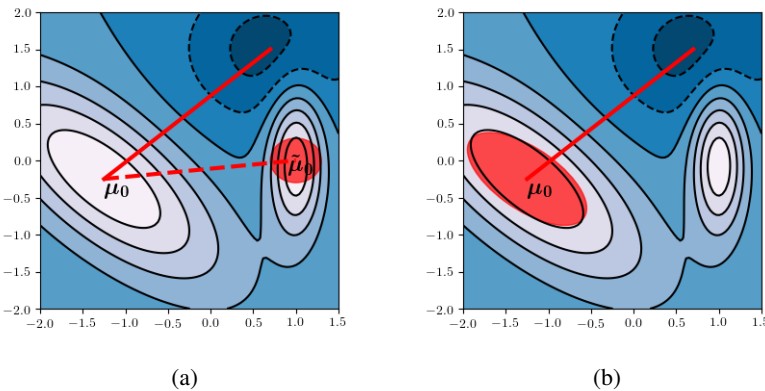

(a)                                           (b)

Figure 1: **The importance of retaining the original minimum**: A number of works have shown the existence of an empirical correlation between generalization error and flatness in the loss landscape around the deep neural network minimum $\boldsymbol{\mu}_0$. Recently popular PAC-Bayes analyses can be seen as a formal way of quantifying this flatness. However, moving forward from simple empirical correlations, existing optimization based non-vacuous bounds compute implicitly a different minimum $\tilde{\boldsymbol{\mu}}_0$ and then evaluate the flatness around that minimum. By contrast we aim to estimate the flatness and a related optimal posterior distribution around the original minimum $\boldsymbol{\mu}_0$.

bounds for a simplified Mnist dataset Dziugaite & Roy (2017) and the Imagenet dataset respectively Zhou et al. (2018). This stands in stark contrast with previous works Bartlett et al. (2017) Neyshabur et al. (2017b) Golowich et al. (2017) which yield bounds that are vacuous by several orders of magnitude and are usually motivated by simple empirical correlation with the generalization error.

Both works rely crucially on applying the PAC-Bayes framework McAllester (1999) which typically addresses the generalization error of stochastic classifiers. Given the randomized empirical error $\mathbb{E}_{\boldsymbol{\theta} \sim Q}[\hat{L}(\boldsymbol{\theta})]$, the randomized population error $\mathbb{E}_{\boldsymbol{\theta} \sim Q}[L(\boldsymbol{\theta})]$, a prior weight distribution $P$, a posterior weight distribution $Q$ and $N$ training samples the PAC-Bayes bound gives a guarantee of the form

$$\mathbb{E}_{\boldsymbol{\theta} \sim Q}[L(\boldsymbol{\theta})] \leq \mathbb{E}_{\boldsymbol{\theta} \sim Q}[\hat{L}(\boldsymbol{\theta})] + A\sqrt{(\mathrm{KL}(Q||P) + B)/N}, \tag{1}$$

with some probability $1 - \delta$ where $A$ and $B$ are constants related to the derivation. The PAC-Bayes bound models the complexity of a classifier as the KL-Divergence between a prior $P$ and a posterior $Q$ weight distribution. Apart from yielding state of the art bounds for SVMs, a further motivation for this framework, is it's Bayesian nature i.e. is the existance of the prior $P$. Complexity is not measured with respect to an arbitrary reference point, but as in the luckiness framework Shawe-Taylor et al. (1998), with respect to a reference point that can potentially incorporate our prior knowledge about good solutions to the classification problem, and can lead to possibly tighter bounds.

In Dziugaite & Roy (2017) the authors model the weights as originating from a Gaussian posterior distribution with diagonal covariance $\boldsymbol{\theta} = \boldsymbol{\mu} + \boldsymbol{\xi} \odot \boldsymbol{\sigma}$ where $\boldsymbol{\xi} \sim \mathcal{N}(0, I)$. Then they optimize directly the stochastic objective resulting from the PAC-Bayes bound, by approximating stochastic quantities with MC sampling. Similarly in Zhou et al. (2018) the authors first compress a DNN with an off the shelf compession algorithm, removing redundant parameters and applying the PAC-Bayes approach to the remaining weights. This can be seen as explicitly minimizing the length of a code describing the DNN. While compression in Zhou et al. (2018) is done explicitly, Dziugaite & Roy (2017) can also be seen under this light, the random variables form a *variational code* whose length is explicitly minimized Blier & Ollivier (2018).

While these two works result in non-vacuous bounds they have a number of important limitations.

- Importantly they provide generalization error guarantees for a *different* classifier than the original. Compressing the neural network or modelling it as originating from a Gaussian distribution whose mean is modified, results in finding a completely different point in the parameter space than the original. The function that the new weights describe might or might not be close to the original. This might seem like an academic problem. However we argue that analyzing networks that typically result from vanilla SGD is equally important potentially leading to better initialization and regularisation of SGD, as well as discovering inherent limitions of our generalization proof toolbox.

- The bounds provided by Dziugaite & Roy (2017)Zhou et al. (2018) are non-vacuous but loose. What is the source of this looseness? Both methods rely in non-convex optimisation that might have simply not converged properly. This is a particular problem in Dziugaite & Roy (2017) where it is well known that VI techniques require laborious hyperparameter tuning, something that has hindered significantly the wide applicability of Bayesian neural networks Wu et al. (2018). Even when care is taken the resuls are widely considered suboptimal in terms of uncertainty estimation for prediction tasks and code length description of the DNN Wu et al. (2018)Blier & Ollivier (2018). As such we argue that one stands to benefit from circumventing these non-convex optimisation procedures when possible.

The PAC-Bayes bound can also be seen under the light of flat minima. Flat minima have been empirically shown to correlate with better generalization Neyshabur et al. (2017a)Keskar et al. (2016). The PAC-Bayes bound can be interpreted as balancing two terms, the randommized empirical loss and the KL complexity term Neyshabur et al. (2017a). As the posterior can be arbitrarily chosen increasing the variance of the noise added to the parameters will typically decrease the KL term while increasing the empirical loss. If the neural network solution corresponds to a flat minimum,

more noise can be added to the parameters without affecting the empirical performance Neyshabur et al. (2017a).

In this work we propose to tackle the above problems by using a second order Taylor expansion of the loss around the minimum, in conjunction with the PAC-Bayes framework. The second order term parameterized by the Hessian matrix corresponds to the curvature of the loss around the minimum. In line with the works linking flat minima to better generalization Neyshabur et al. (2017a)Keskar et al. (2016), we are effectively estimating the flatness of the minimum along all parameter dierections. Thus we are able to add more noise to the flat directions and less noise to the curved ones, something that will typically decrease significantly the KL term in the PAC-Bayes framework while ensuring that the loss of the stochastic classifier remains small. Crucially for the case of Gaussian posteriors the optimal posterior covariance can be found in closed form, allowing us to circumvent a non-convex optimization procedure by incuring an approximation penalty due to using a second order approximation of the loss.

The resulting approximation is closely linked to the *Laplace* approximation in Bayesian statistics Bishop (2006). It and similar approximations to the posterior have a appeared a number of times in the literature of bayesian neural networks yielding good results in a number of tasks Maddox et al. (2019); Khan et al. (2019); Ritter et al. (2018); Zhang et al. (2017); Khan et al. (2018). Similarly second order approximations of the loss around a minimum have a long history in the literature of DNN compression Dong et al. (2017); Wang et al. (2019); Peng et al. (2019); LeCun et al. (1990); Hassibi & Stork (1993) often yielding state of the art results in parameter reduction.

Even though the resulting posterior is optimal with respect to our approximation, being able to chose a data driven informative prior still leaves room for looseness. Importantly the prior in the PAC-Bayes framework can depend on the data generating distribution but not on the training set used to train the evaluated classifier. Workarounds include choosing data driven priors by training a separate classifier on a different training set, or using the same training set but enforcing that the training set is not too informative about each individual training signal. The later can be formalized through the framework of differential privacy. Both of the above will usually involve some non-convex optimisation procedure leaving again doubt as to the optimality of the classifier complexity estimate. We show that for the case of Gaussian priors and posteriors with diagonal covariance we can derive the optimal *invalid* prior covariance in closed form.

The invalid prior cannot be used to prove generalization, however it can be used as a sanity check to see whether proving generalization is possible in principle. The optimal solution with respect to both prior and posterior covariance results in a lower bound on a function closely related to the PAC-Bayes bound (but not exactly equal given that we make a number of approximations). Through experiments we find that, depending on the hardness of the dataset, one can find cases where the set of feasible solutions implied from the lower bound and the set of non-vacuous PAC-Bayes bound solutions don't intersect. One is unable to prove generalization, even through choosing a prior in an invalid manner.

A number of works Achille & Soatto (2018); Dziugaite & Roy (2017); Germain et al. (2016) have noted the similarity between the stochastic PAC-Bayes objective and the objective

$$C_\beta(\mathcal{D}; P, Q) = \mathbb{E}_{\boldsymbol{\theta} \sim Q}[\hat{L}(\boldsymbol{\theta})] + \beta \mathrm{KL}(Q||P). \qquad (2)$$

For $\beta = 1$ this is known as the Evidence Lower Bound (ELBO) objective in Variational Inference literature Kingma et al. (2015)Bishop (2006). In the Information Bottleneck framework Achille & Soatto (2018)Tishby et al. (2000) it is know as the IB-Lagrangian. More recently the same objective has been interpreted as the "task complexity" Achille et al. (2019). Then $\beta$ has the role of regulating the amount of information in the randomized neural network Achille & Soatto (2018), smaller values correspond to more information and potential to overfit. While the objectives 1 and 2 are not entirely equivalent due to a square root term over the KL divergence in the PAC-Bayes case, we will be using the IB formulation as removing the square root will ease our derivations. We note that Dziugaite & Roy (2017) have used the two objectives intercheangably with no significant difference in results.

A number of preprints have appeared on Arxiv trying to link PAC-Bayes to flat minima Wang et al. (2018); Li et al. (2019); Tsuzuku et al. (2019); Yang et al. (2019). These typically involve heuristic choices for the optimal posterior, do not analyze the role of the prior, and focus on quantities that simply correlate with the generalization error. Motivating vacuous generalization bounds on the

basis of empirical correlations with generalization error has been criticised in a number of recent works Kawaguchi et al. (2017); Nagarajan & Kolter (2019b); Pitas et al. (2019).

## 2 CONTRIBUTIONS

- We propose to analyze a PAC-Bayes bound through a second order Taylor expansion of the randomized empirical loss in the closely related IB-Lagrangian objective. We thus pay an error resulting from the second order approximation in order to circumvent non-convex optimisation procedures, which might require extensive hyperparameter tuning and not converge properly.

- We are able to find a lower bound to this second order taylor expansion that corresponds to an invalid PAC-Bayes prior (the prior is training set *dependent*). Experimentally we find cases where the set of feasible solutions implied by the lower bound does not intersect with the set of non-vacuous PAC-Bayes accuracy-complexity pairs. This in turn implies easy and hard cases were one can and cannot prove generalization using Gaussian priors and posteriors with diagonal covariance.

- While we rely on approximations, we see empirically that in easy cases one can find non-vacuous generalization bounds using a very simple baseline where the prior is crucially centered at the random DNN initialization. This in turn highlights the insightful prior mean choice as the main innovation in previous non-vacous bound works Dziugaite & Roy (2017) with bound optimization through SGD having a complementary role.

- We motivate a layerwise method to optimise PAC-Bayes bounds with respect to the posterior variance, reducing the required computations significantly.

## 3 PAC-BAYESIAN FRAMEWORK

We consider the hypothesis class $\mathcal{H}_L$ realized by the feedforward neural network architecture of depth $L$ with coordinate-wise activation functions $\sigma$ defined as the set of functions $f_{\boldsymbol{\theta}} : \mathcal{X} \to \mathcal{Y}$ ($\mathcal{X} \subseteq \mathbb{R}^p, \mathcal{Y} \subseteq \mathbb{R}^K$) with $f_{\boldsymbol{\theta}}(\boldsymbol{x}) = \sigma(\sigma(...\sigma(\boldsymbol{x}^T\mathbf{W}_0)\mathbf{W}_1)\mathbf{W}_2)..)\mathbf{W}_L)$ where $\boldsymbol{\theta} \in \Theta_L \subseteq \mathbb{R}^d$ is a vectorization of the weights and $\Theta_L = \mathbb{R}^{p \times k_1} \times \mathbb{R}^{k_1 \times k_2} \times ... \times \mathbb{R}^{k_L \times K}$. Given the loss function $\ell(\cdot, \cdot)$ we can define the population loss: $L(\boldsymbol{\theta}) := \mathbb{E}_{(\boldsymbol{x},\boldsymbol{y}) \sim \mathcal{P}} \ell(f_{\boldsymbol{\theta}}(\boldsymbol{x}), \boldsymbol{y})$ and given a training set of $N$ instances $\mathcal{D} = \{(\boldsymbol{x}_j, \boldsymbol{y}_j)\}_{j=1}^N$ the empirical loss $\hat{L}(\boldsymbol{\theta}) := \frac{1}{N} \sum_{i=1}^N \ell(f_{\boldsymbol{\theta}}(\boldsymbol{x}_i), \boldsymbol{y}_i)$.

The PAC-Bayesian framework McAllester (1999) provides generalization error guarantees for randomized classifiers drawn from a posterior distribution $Q$. We will use the following form of the PAC-Bayes bound.

**Theorem 3.1.** *(PAC-Bayesian theorem McAllester (1999)) For any data distribution over $\mathcal{X} \times \mathcal{Y}$, we have that the following bound holds with probability at least $1 - \delta$ over random i.i.d. samples $\mathcal{D} = \{(\boldsymbol{x}_j, \boldsymbol{y}_j)\}_{j=1}^N$ of size $N$ drawn form the data distribution:*

$$\mathbb{E}_{\boldsymbol{\theta} \sim Q}[L(\boldsymbol{\theta})] \leq \mathbb{E}_{\boldsymbol{\theta} \sim Q}[\hat{L}(\boldsymbol{\theta})] + \sqrt{\frac{\text{KL}(Q||P) + \ln \frac{2(N-1)}{\delta}}{2N}}. \tag{3}$$

*Here $Q$ is an arbitrary "posterior" distribution over parameter vectors, which may depend on the sample $\mathcal{D}$ and on the prior $P$.*

The framework models the complexity of the randomized classifier as the KL-Divergence between the posterior $Q$ and a prior $P$. The prior $P$ must be valid in that it cannot depend in any way on the training data. On the contrary the posterior $Q$ can be chosen to be any arbitrary distribution. This flexibility allows one to model deterministic neural networks as the mean of an arbitrary posterior distribution, thus deriving results for a stochastic but closely related classifier.

The square root in 3 will make subsequent calculations cumbersome. We will therefore be analyzing the similar objective

$$C_\beta(\mathcal{D}; P, Q) = \mathbb{E}_{\boldsymbol{\theta} \sim Q}[\hat{L}(\boldsymbol{\theta})] + \beta \text{KL}(Q||P), \tag{4}$$

which is known as the IB-Lagrangian Achille & Soatto (2018); Tishby et al. (2000). While the two objectives are not exactly equivalent, we argue through experiments that the differences are minimal and our results translate to the original PAC-Bayes bound 3.

## 4 QUADRATIC APPROXIMATION AND LOWER BOUND

The stochastic and non-convex objective 4 is difficult to analyze theoretically. As such we first propose to expand the randomized loss using a Taylor expansion which will make the subsequent analysis tractable. We get

$$
\begin{aligned}
C_\beta(\mathcal{D}; P, Q) &= \mathop{\mathbb{E}}_{\boldsymbol{\theta} \sim Q}[\hat{L}(\boldsymbol{\theta})] + \beta \mathrm{KL}(Q||P) \\
&\approx \mathop{\mathbb{E}}_{\boldsymbol{\eta} \sim Q'}[\left(\frac{\partial \hat{L}(\boldsymbol{\theta})}{\partial \boldsymbol{\theta}}\right)^T \boldsymbol{\eta} + \frac{1}{2}\boldsymbol{\eta}^T \mathbf{H} \boldsymbol{\eta} + O(||\boldsymbol{\eta}||^3)] + \beta \mathrm{KL}(Q||P) \\
&\leq \mathop{\mathbb{E}}_{\boldsymbol{\eta} \sim Q'}[\frac{1}{2}\boldsymbol{\eta}^T \mathbf{H} \boldsymbol{\eta}] + \beta \mathrm{KL}(Q||P),
\end{aligned}
\tag{5}
$$

where $Q'$ is a centered version of $Q$. We've made a number of assumptions. In the second line we assumed that the loss $\hat{L}(\boldsymbol{\theta})$ at the minimum is 0. In line 3 we assumed that the gradient $\partial \hat{L}(\boldsymbol{\theta})/\partial \boldsymbol{\theta}$ at the minimum is also 0, the term $O(||\boldsymbol{\eta}||^3)$ is negligible and that the loss is locally convex resulting in the quadratic approximation being an upper bound. For a well trained overparametrized DNN assuming that the loss and the gradient at the minimum are zero is reasonable. Assuming local convexity at the minimum is also often a reasonable assumption Sagun et al. (2017); Li et al. (2018). Regarding the validity of the quadratic approximation we note that it often provides state of the art results in DNN compression Dong et al. (2017); Wang et al. (2019); Peng et al. (2019); LeCun et al. (1990); Hassibi & Stork (1993) being highly informative about parameter relevence. Finally we will be analyzing Gaussian posterior distributions with diagonal or close to diagonal covariance which can be assumed to sufficiently concentrate around the minimum making the local second order approximation meaningful.

### 4.1 OPTIMAL POSTERIOR

We make the following modeling choices $Q = \mathcal{N}(\boldsymbol{\mu}_0, \boldsymbol{\Sigma}_0)$ and $P = \mathcal{N}(\boldsymbol{\mu}_1, \lambda\boldsymbol{\Sigma}_1)$ which are popular in VI and PAC-Bayes literature. We can then show that the optimal posterior covariance of the above objective for fixed prior and posterior means has a closed form solution.

**Lemma 4.1.** *The convex optimization problem* $\min_{\boldsymbol{\Sigma}_0} \mathop{\mathbb{E}}_{\boldsymbol{\eta} \sim Q'}[\frac{1}{2}\boldsymbol{\eta}^T \mathbf{H}_l \boldsymbol{\eta}] + \beta \mathrm{KL}(Q||P)$ *where* $Q = \mathcal{N}(\boldsymbol{\mu}_0, \boldsymbol{\Sigma}_0)$ *and* $P = \mathcal{N}(\boldsymbol{\mu}_1, \lambda\boldsymbol{\Sigma}_1)$ *has the global minimum:*

$$
\boldsymbol{\Sigma}_0^* = \beta(\mathbf{H}_l + \frac{\beta}{\lambda}\boldsymbol{\Sigma}_1^{-1})^{-1},
\tag{6}
$$

*where* $\mathbf{H}_l$ *captures the curvature in the directions of the parameters, while* $\boldsymbol{\Sigma}_1$ *is a chosen prior covariance.*

This can been seen as a *Laplace* approximation Bishop (2006) to the posterior around the MAP solution. In practice we will be using 6 by performing a grid search over the parameters $\beta$ and $\lambda$. Then we will try to find Pareto optimal pairs balancing the accuracy of the randomized classifier and the KL complexity term.

### 4.2 OPTIMAL PRIOR

The above solution is not optimal with respect to the *prior* covariance in that we have up to now chosen it arbitrarily. Furthermore given that the choice of the random initialization as the prior mean has been independently shown to result in much tighter bounds in a variety of settings Dziugaite & Roy (2017); Nagarajan & Kolter (2019a) one would wish to isolate the effects of the prior mean on the bound tightness from the prior covariance.

PAC-Bayesian theory allows one to choose an informative prior, however the prior can only depend on the data generating distribution and *not* the training set. A number of previous works Parrado-Hernández et al. (2012); Catoni (2003); Ambroladze et al. (2007) have used this insight mainly on simpler linear settings and usually by training a classifier on a separate training set and using the

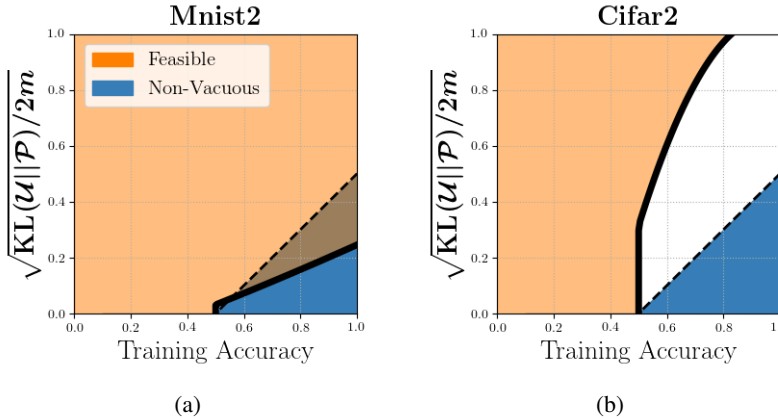

(a)  (b)

Figure 2: **Feasible solutions vs Non-Vacuous solutions**: We merge the 10 classes in Mnist and Cifar to create simpler 2 class problems. For different values of $\beta$ we compute the optimal complexity terms $\sqrt{\frac{\text{KL}(Q||P)+\ln\frac{2(N-1)}{\delta}}{2N}}$ using 7, 8. We compute the accuracy of the stochastic classifier with MC for 5 samples. We plot this lower bound with the solid black line. All points above it are feasible. We see that for the Mnist problem the two regions intersect suggesting that we might be able to prove generalization using Gaussians with diagonal covariance. By contrast in the Cifar case the two regions do not intersect suggesting that the prior and posterior means have a prohibitive distance between them, and we cannot prove generalization with diagonal covariances.

result as a prior. Recently Dziugaite & Roy (2018) have proposed to use the original training set to derive valid priors by imposing differential privacy constraints.

We ignore these concerns for the moment and optimize the prior covariance directly. The objective is non-convex, however for the case of diagonal prior and posterior covariances we can find the global minimum.

**Lemma 4.2.** *The optimal prior and posterior for* $C_\beta(\mathcal{D}; P, Q) = \underset{\boldsymbol{\eta} \sim Q'}{\mathbb{E}}[\frac{1}{2}\boldsymbol{\eta}^T \mathbf{H}_l \boldsymbol{\eta}] + \beta \text{KL}(Q||P)$ *with* $Q = \mathcal{N}(\boldsymbol{\mu}_0, \boldsymbol{\Sigma}_0)$ *and* $P = \mathcal{N}(\boldsymbol{\mu}_1, \lambda \boldsymbol{\Sigma}_1)$ *and assuming that* $\boldsymbol{\Sigma}_1^{-1} = \boldsymbol{\Lambda}_1 = \text{diag}(\Lambda_{11}, \Lambda_{21}, ..., \Lambda_{k1})$ *and* $\mathbf{H}_l = \text{diag}(h_{1l}, h_{2l}, ..., h_{kl})$ *have:*

$$\Lambda_{i1}^* = \frac{\lambda}{2\beta}[\sqrt{h_{il}^2 + \frac{4\beta h_{il}}{(\mu_{i0} - \mu_{i1})^2}} - h_{il}], \tag{7}$$

$$\Lambda_{i0}^* = \frac{1}{2\beta}[h_{il} + \sqrt{h_{il}^2 + \frac{4\beta h_{il}}{(\mu_{i0} - \mu_{i1})^2}}]. \tag{8}$$

*where* $\mathbf{H}_l$ *encodes the local curvature at the the minimum,* $\boldsymbol{\mu}_1$ *corresponds to the random initialization (by design) of the DNN, and* $\boldsymbol{\mu}_0$ *corresponds to the minimum obtained after optimization.*

*For our choice of Gaussian prior and posterior, the following is a lower bound on the IB-Lagrangian under any Gaussian prior covariance:*

$$\min_{\boldsymbol{\Sigma}_0, \boldsymbol{\Sigma}_1} C_\beta(\mathcal{D}; P, Q) \gtrsim \frac{1}{2}(\sum_i a_{il}(\mu_{i0} - \mu_{i1})^2 + \beta \sum_i \ln(\frac{h_{il} + a_{il}}{a_{il}})), \tag{9}$$

*where* $a_{il} \triangleq a_{il}(\beta, \mu_{i0}, \mu_{i1}, h_{il}) = \frac{1}{2}[\sqrt{h_{il}^2 + \frac{4\beta h_{il}}{(\mu_{i0} - \mu_{i1})^2}} - h_{il}]$.

The above result is intuitively pleasing setting a lower bound to what we can achieve which depends only on the initialization (by design), obtained minimum, curvature at the minimum and the regularization parameter $\beta$. In particular the scaling factor $\lambda$ has disappeared.

We now make some important notes about what one can and *cannot* prove using these results, by stressing that the above result is a necessary but not a *sufficient* condition for generalization under our prior and posterior modeling.

- Given a deterministic deep neural network and it's initialization (or other prior mean) one *can* rule out being able to prove generalization using any choice of diagonal covariances when modeling the priors and posteriors as multivariate Gaussians with fixed means. Modeling with other distributions may give different results[†].

- One *cannot* prove generalization using this result, even in the case when the prior mean is valid (only distribution dependent) and the feasible and non-vacuous sets intersect. One still has to compute the prior and posterior covariances in a valid manner. As such a computationally feasible region given finite data and computational resources as well as privacy constraints, might be much smaller than the one we derive here.

We plot our lower bound for simplified 2 class Mnist and Cifar problems in Figure 2. We see that while for the Mnist problem the feasible and non-vacuous regions intersect the same is not true for the Cifar problem. What remains is to see if our results apply for the case of valid priors. First we detail a number of computational issues in section 5.

## 5 COMPUTATIONAL ASPECTS

We now present a number of computational and memory issues associated with the Hessian of a modern deep neural network. There is ambiguity in the literature about the size of the Hessians that can be *computed* exactly Kunstner et al. (2019). There have been few results in this area and the main problem seems to be that the relevant computations are not well supported from common auto-differentiation libraries, such as Tensorflow and Keras. However there is certainty on the fact that storing and manipulating the full Hessian of a number of modern deep neural network architectures would be infeasible as the matrix is of size $\mathbf{H} \in \mathbb{R}^{d \times d}$ where $d$ is the number of parameters. As a point of reference a dense uncompressed Numpy matrix for $d = 50000$ takes up 20GB of memory. As such we detail in the next section a number of approximations that make both computing and storing the Hessian feasible.

### 5.1 APPROXIMATING THE FULL HESSIAN

As noted in Kunstner et al. (2019) the generalized Gauss-Newton approximation of the Hessian $\mathbf{H}(\boldsymbol{\theta})$ coincides with the Fisher matrix $\mathbf{F}(\boldsymbol{\theta}) = \sum_n \mathbb{E}_{p_{\boldsymbol{\theta}}(y|x_n)}[\nabla_{\boldsymbol{\theta}} \log p_{\boldsymbol{\theta}}(y|x_n) \nabla_{\boldsymbol{\theta}} \log p_{\boldsymbol{\theta}}(y|x_n)^{\mathrm{T}}]$ in modern deep learning architectures. While the Fisher matrix is difficult to compute exactly one can compute an unbiased but noisy estimate as Martens & Grosse (2015)

$$\mathbf{F}(\boldsymbol{\theta}) \approx \sum_n [\nabla_{\boldsymbol{\theta}} \log p_{\boldsymbol{\theta}}(\tilde{y}_n|x_n) \nabla_{\boldsymbol{\theta}} \log p_{\boldsymbol{\theta}}(\tilde{y}_n|x_n)^{\mathrm{T}}], \tag{10}$$

where care must be taken to sample $\tilde{y}_n$ from the model predictive distribution $\tilde{y}_n \sim p_{\boldsymbol{\theta}}(y|x_n)$. Additionally we note that the interpretation of the outputs after the softmax as probabilities is not well grounded theoretically Gal & Ghahramani (2016). Determining the true predictive distribution requires MC sampling for example by taking multiple dropout samples Gal & Ghahramani (2016).

We now make two additional notes regarding computational aspects of the above. The approximation of the Hessian can be computed efficiently as the outer product of large but manageable gradient vectors. The main computational burden after we approximate the Hessian, and given that we choose a standard normal prior, is inverting a matrix of the form $\tilde{\mathbf{H}} + \alpha I$. This problem can be tackled in a few different ways. The simplest would be to consider only the diagonal elements of $\tilde{\mathbf{H}}$ and the resulting diagonal matrix can be efficiently inverted. However inversion of the full matrix $\tilde{\mathbf{H}} + \alpha I$ is also possible recursively using the Sherman-Morrison formula.

---

[†]While our result is a formal lower bound on what is achievable by 6, it's applicability on direct minimization of the IB-Lagrangian 4 depends on the tightness of the second order approximation.

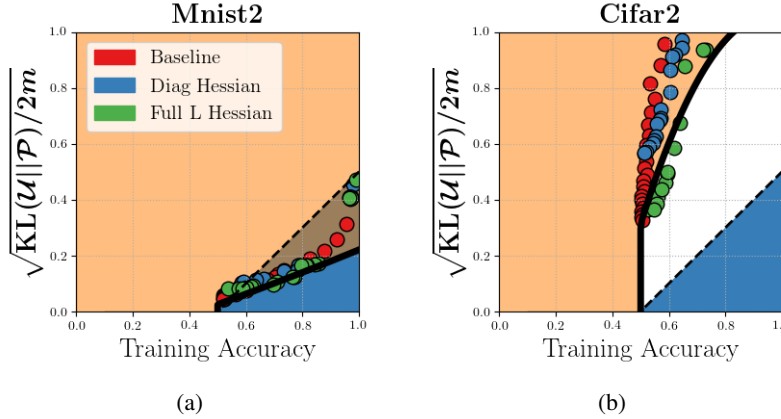

(a)                   (b)

Figure 3: **Accuracy vs Complexity for different bounds**: We plot $\sqrt{\frac{KL(Q||P)+\ln\frac{2(N-1)}{\delta}}{2N}}$ and training accuracy (of the randomized classifier) for different architectures and datasets. Points to the right of the dashed line correspond to non-vacuous pairs. All Mnist bounds are non-vacuous. All Cifar bounds are vacuous. We are able to progressively get tighter bounds by using the diagonal Hessian and then the full layerwise Hessian. The improvement is larger over the more difficult Cifar dataset.

Further issues exist with computing the KL-Divergence of large multivariate Gaussians with non-diagonal covariances in closed form which includes a determinant term that has to be computed with the matrix determinant lemma, as well as sampling efficiently from these distributions. As such we have used the diagonal variant of approximation 10 for our lower bound, but perform a layerwise approximation of the Hessian for all other experiments. We detail this layerwise approximation in the next section, and motivate it theoretically.

## 5.2 LAYERWISE APPROXIMATION

For a specific case of the empirical loss we will now derive an upper bound on 4 which is more suitable for optimization.

**Lemma 5.1.** *Assuming the following empirical loss $\hat{L}(\boldsymbol{\theta}) = ||f_{\boldsymbol{\theta}}(\mathbf{X}) - \mathbf{Y}||_F$ with $\mathbf{X} = [\boldsymbol{x}_0, ..., \boldsymbol{x}_N]$ and $\mathbf{Y} = [\boldsymbol{y}_0, ..., \boldsymbol{y}_N]$ the following is an upper bound on the IB Lagrangian given that we are at a local minimum:*

$$C_\beta(\mathcal{D}; P, Q) \lesssim \sum_l \sqrt{\sum_j c_{lj} \mathop{\mathbb{E}}_{\boldsymbol{\eta} \sim Q'_{lj}} [\frac{1}{2}\boldsymbol{\eta}^T \mathbf{H}_{lj}\boldsymbol{\eta}]} + \beta \sum_{l,j} \mathrm{KL}(Q_{lj}||P_{lj}), \tag{11}$$

*where $l$ denotes different layers, $j$ denotes the different neurons at each layer (we assume the same number for simplicity), $\mathbf{H}_{lj}$ denotes the local Hessian, and $Q'_{lj}$ is a centered version of $Q_{lj}$. The local Hessian can be computed efficiently as $\mathbf{H}_{lj} = \frac{1}{N}\sum_{i=1}^N \boldsymbol{z}_{l-1}^i {\boldsymbol{z}_{l-1}^i}^T$ and $\boldsymbol{z}_{l-1}^i$ is the latent representation input to layer $l$ for signal $i$.*

We see that we have managed to upper bound the empirical randomized loss by a scaled sum of quadratic terms involving layerwise Hessian matrices and centered random noise vectors. Intuitively we have reduced the complexity of our optimization problem simply by turning it into a number of separate subproblems. The local Hessians can be computed efficiently from outer products of a forward pass of the dataset. Apart from avoiding using backpropagation, breaking the Hessian into subproblems in this manner allows us to move beyond the simplistic diagonal approximation. Implicitly the Hessian now has a block diagonal structure and the blocks are small enough to be inverted directly for the architectures used in this paper. For architectures with larger latent representations the Sherman-Morrison formula can be used instead. While 5.1 holds for a specific loss which is uncommon in practice, we have found empirically that solving the posterior optimization problem in a layerwise manner gives good results with the more common loss functions.

## 6 EXPERIMENTS

We now make a number of experiments on the simplified 2 class Mnist and Cifar datasets. Specifically we test the architecture

$$\text{input} \rightarrow 300\text{FC} \rightarrow 300\text{FC} \rightarrow \#classes\text{FC} \rightarrow \text{output}$$

on Mnist and

$$\text{input} \rightarrow 200\text{FC} \rightarrow 200\text{FC} \rightarrow \#classes\text{FC} \rightarrow \text{output}$$

on Cifar. We note that the Mnist architecture corresponds exactly to the architecture T-300$^2$ p.7 in Dziugaite & Roy (2017). We train each configuration to 100% accuracy and derive the layerwise Hessians. We model the prior and posteriors as multivariate Gaussians centered at the initialization and deterministic solution respectively. For the prior we choose the uninformative unit diagonal covariance, scaled by the free parameter $\lambda$. The baseline posterior that we use has the same diagonal covariance as the prior. For the baseline we perform a grid search over $\lambda$ which increases the complexity negligibly. For the optimized posterior we initially test a diagonal approximation of the Hessian "Diag Hessian" which results in an optimal diagonal covariance. We perform a grid search for the parameters $\lambda$ and $\beta$ using formula 6 to derive candidates for the optimal posterior covariance. For each point on the grid we calculate the empirical accuracy over the training set using Monte Carlo sampling and 5 samples, as well as the complexity term $\sqrt{\frac{\text{KL}(Q||P)+\ln \frac{2(N-1)}{\delta}}{2N}}$. We then choose the Pareto optimal points from all candidates. We plot the results in Figure 3.

Interestingly we see that for the case of Mnist the baseline is tight with respect to our lower bound and provides non-vacuous bounds. Therefore not much improvement can be achieved using the Hessian approach. This implies a more careful interpretation of the results in Dziugaite & Roy (2017). We see that non-vacuity can also be achieved as a result of the problem being very simple, and the choice of the prior mean being the random initialization. The optimization techniques employed in Dziugaite & Roy (2017) should simply tighten the bound further, mainly by moving the posterior mean closer to the prior mean (the random initialization). For the case of Cifar we see that we can significantly tighten the bound. However we cannot manage to turn a vacuous bound to a non-vacuous one in line with our lower bound. We also test a block-diagonal approximation to the Hessian "Full L Hessian" where each neuron of the network has it's own block. Our non-diagonal layerwise approximation however crude seems to improve significantly over the diagonal case and slightly crosses our diagonal lower bound. This suggests that better approximations of the Hessian as well as better prior means apart from the random initialization might be needed to prove generalization in complex datasets and architectures.

## 7 CONCLUSION

We have presented a lower bound on an approximation of the IB-Lagrangian for the case of multivariate Gaussian priors and posteriors with diagonal covariance. This coincides with a lower bound on a PAC-Bayesian generalization bound for an invalid (training set dependent) prior. For cases where the feasible and non-vacuous regions intersect we have seen that it is possible to reach the lower bound and achieve non-vacuous bounds by using valid non-informative priors. We have also presented closed form solutions for the optimal posteriors given fixed means under our modeling assumptions, and motivated theoretically breaking the estimation into layerwise subproblems. Crucially all results depend on high quality estimates of the Hessian which remains an open topic of research for large scale modern deep neural networks.

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

# APPENDIX

## A. ADDITIONAL EXPERIMENTS

We test the architectures

$$\text{input} \rightarrow 300\text{FC} \rightarrow 300\text{FC} \rightarrow \#classes\text{FC} \rightarrow \text{output}$$

on Mnist and

$$\text{input} \rightarrow 200\text{FC} \rightarrow 200\text{FC} \rightarrow \#classes\text{FC} \rightarrow \text{output}$$

on Cifar. We conduct additional experiments on the original Cifar10 and Mnist10 datasets, as well as Cifar5 and Mnist5 where we merge the 10 classes into 5. The results are consistent across datasets, with more improvement when incorporating the Hessian for more difficult datasets.

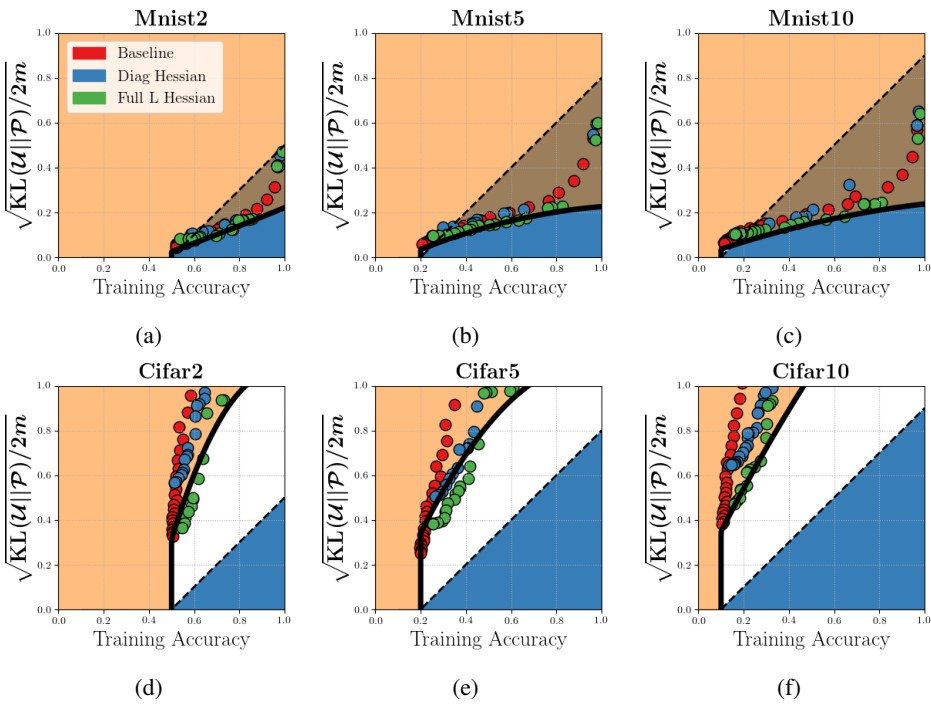

Figure 4: **Accuracy vs Complexity for different bounds**: We plot $\sqrt{\frac{\text{KL}(Q||P) + \ln\frac{2(N-1)}{\delta}}{2N}}$ and training accuracy (of the randomized classifier) for different architectures and datasets. Points to the right of the dashed line correspond to non-vacuous pairs. All Mnist bounds are non-vacuous. All Cifar bounds are vacuous. We are able to progressively get tighter bounds by using the diagonal Hessian and then the full layerwise Hessian. The improvement is larger over the more difficult Cifar dataset.

## B. PROOFS

**Lemma 4.1.** The convex optimization problem $\min_{\boldsymbol{\Sigma}_0} \mathbb{E}_{\boldsymbol{\eta} \sim Q'}[\frac{1}{2}\boldsymbol{\eta}^T\mathbf{H}_l\boldsymbol{\eta}] + \beta \mathrm{KL}(Q||P)$ where $Q = \mathcal{N}(\boldsymbol{\mu}_0, \boldsymbol{\Sigma}_0)$ and $P = \mathcal{N}(\boldsymbol{\mu}_1, \lambda\boldsymbol{\Sigma}_1)$ has the global minimum:

$$\boldsymbol{\Sigma}_0^* = \beta(\mathbf{H}_l + \frac{\beta}{\lambda}\boldsymbol{\Sigma}_1^{-1})^{-1}, \tag{12}$$

where $\mathbf{H}_l$ captures the curvature in the directions of the parameters, while $\boldsymbol{\Sigma}_1$ is a chosen prior covariance.

*Proof.*

$$
\begin{aligned}
C_\beta(\mathcal{D}; P, Q) &= \mathbb{E}_{\boldsymbol{\eta} \sim Q'}[\frac{1}{2}\boldsymbol{\eta}^T\mathbf{H}_l\boldsymbol{\eta}] + \beta \mathrm{KL}(Q||P) = \\
&\quad \mathbb{E}_{\boldsymbol{\eta} \sim Q'}[\frac{1}{2}\mathrm{tr}(\mathbf{H}_l\boldsymbol{\eta}\boldsymbol{\eta}^T)] + \beta \mathrm{KL}(Q||P) = \\
&\quad \frac{1}{2}\mathrm{tr}(\mathbf{H}_l \mathbb{E}_{\boldsymbol{\eta} \sim Q'}[\boldsymbol{\eta}\boldsymbol{\eta}^T]) + \beta \mathrm{KL}(Q||P) = \\
&\quad \frac{1}{2}\mathrm{tr}(\mathbf{H}_l\boldsymbol{\Sigma}_0) + \frac{\beta}{2}(\mathrm{tr}(\frac{1}{\lambda}\boldsymbol{\Sigma}_1^{-1}\boldsymbol{\Sigma}_0) - k + \frac{1}{\lambda}(\boldsymbol{\mu}_0 - \boldsymbol{\mu}_1)^{\mathrm{T}}\boldsymbol{\Sigma}_1^{-1}(\boldsymbol{\mu}_0 - \boldsymbol{\mu}_1) \\
&\quad + \ln\left(\frac{\det \lambda\boldsymbol{\Sigma}_1}{\det \boldsymbol{\Sigma}_0}\right))
\end{aligned}
\tag{13}
$$

The gradient with respect to $\boldsymbol{\Sigma}_0$ is

$$\frac{\partial C_\beta(\mathcal{D}; P, Q)}{\partial \boldsymbol{\Sigma}_0} = [\frac{1}{2}\mathbf{H}_l + \frac{\beta}{2\lambda}\boldsymbol{\Sigma}_1^{-1} - \frac{\beta}{2}\boldsymbol{\Sigma}_0^{-1}]. \tag{14}$$

Setting it to zero, we obtain the minimizer $\boldsymbol{\Sigma}_0^* = \beta(\mathbf{H}_l + \frac{\beta}{\lambda}\boldsymbol{\Sigma}_1^{-1})^{-1}$. $\qquad\square$

**Lemma 4.2.** The optimal prior and posterior for $C_\beta(\mathcal{D}; P, Q) = \mathbb{E}_{\boldsymbol{\eta} \sim Q'}[\frac{1}{2}\boldsymbol{\eta}^T\mathbf{H}_l\boldsymbol{\eta}] + \beta \mathrm{KL}(Q||P)$ with $Q = \mathcal{N}(\boldsymbol{\mu}_0, \boldsymbol{\Sigma}_0)$ and $P = \mathcal{N}(\boldsymbol{\mu}_1, \lambda\boldsymbol{\Sigma}_1)$ and assuming that $\boldsymbol{\Sigma}_1^{-1} = \boldsymbol{\Lambda}_1 = \mathrm{diag}(\Lambda_{11}, \Lambda_{21}, ..., \Lambda_{k1})$ and $\mathbf{H}_l = \mathrm{diag}(h_{1l}, h_{2l}, ..., h_{kl})$ have:

$$\Lambda_{i1}^* = \frac{\lambda}{2\beta}[\sqrt{h_{il}^2 + \frac{4\beta h_{il}}{(\mu_{i0} - \mu_{i1})^2}} - h_{il}], \tag{15}$$

$$\Lambda_{i0}^* = \frac{1}{2\beta}[h_{il} + \sqrt{h_{il}^2 + \frac{4\beta h_{il}}{(\mu_{i0} - \mu_{i1})^2}}]. \tag{16}$$

where $\mathbf{H}_l$ encodes the local curvature at the the minimum, $\boldsymbol{\mu}_1$ corresponds to the random initialization (by design) of the DNN, and $\boldsymbol{\mu}_0$ corresponds to the minimum obtained after optimization.

For our choice of Gaussian prior and posterior, the following is a lower bound on the IB-Lagrangian under any Gaussian prior covariance:

$$\min_{\boldsymbol{\Sigma}_0, \boldsymbol{\Sigma}_1} C_\beta(\mathcal{D}; P, Q) \gtrsim \frac{1}{2}(\sum_i a_{il}(\mu_{i0} - \mu_{i1})^2 + \beta \sum_i \ln(\frac{h_{il} + a_{il}}{a_{il}})), \tag{17}$$

where $a_{il} \triangleq a_{il}(\beta, \mu_{i0}, \mu_{i1}, h_{il}) = \frac{1}{2}[\sqrt{h_{il}^2 + \frac{4\beta h_{il}}{(\mu_{i0} - \mu_{i1})^2}} - h_{il}]$.

*Proof.* Setting $\mathbf{\Lambda}_1 = \mathbf{\Sigma}_1^{-1}$ We can then see that the minimizer is equal to $\mathbf{\Sigma}_0^* = \beta(\mathbf{H}_l + \frac{\beta}{\lambda}\mathbf{\Lambda}_1)^{-1}$. Substituting $\mathbf{\Sigma}_0 = \mathbf{\Sigma}_0^*$ in $C_\beta(\mathcal{D}; P, Q)$ we obtain:

$$
\begin{aligned}
C_\beta(\mathcal{D}; P, Q)|_{\mathbf{\Sigma}_0 = \mathbf{\Sigma}_0^*} = &\underset{\boldsymbol{\eta} \sim Q}{\mathbb{E}}[\frac{1}{2}\boldsymbol{\eta}^T \mathbf{H}_l \boldsymbol{\eta}] + \beta \mathrm{KL}(Q||P)|_{\mathbf{\Sigma}_0 = \mathbf{\Sigma}_0^*} = \\
&\frac{1}{2}\mathrm{tr}(\mathbf{H}_l \beta(\mathbf{H}_l + \frac{\beta}{\lambda}\mathbf{\Lambda}_1)^{-1}) + \frac{\beta}{2}(\mathrm{tr}(\frac{1}{\lambda}\mathbf{\Lambda}_1 \beta(\mathbf{H}_l + \frac{\beta}{\lambda}\mathbf{\Lambda}_1)^{-1}) \\
&+ \frac{1}{\lambda}(\boldsymbol{\mu}_0 - \boldsymbol{\mu}_1)^T \mathbf{\Lambda}_1(\boldsymbol{\mu}_0 - \boldsymbol{\mu}_1) - k + \ln\left(\frac{\det \lambda \mathbf{\Lambda}_1^{-1}}{\det \beta(\mathbf{H}_l + \frac{\beta}{\lambda}\mathbf{\Lambda}_1)^{-1}}\right)) \\
=&\frac{\beta}{2}\mathrm{tr}(\mathbf{H}_l(\mathbf{H}_l + \frac{\beta}{\lambda}\mathbf{\Lambda}_1)^{-1}) + \frac{\beta^2}{2\lambda}(\mathrm{tr}(\mathbf{\Lambda}_1(\mathbf{H}_l + \frac{\beta}{\lambda}\mathbf{\Lambda}_1)^{-1})) \\
&+ \frac{\beta}{2}(+\frac{1}{\lambda}(\boldsymbol{\mu}_0 - \boldsymbol{\mu}_1)^T \mathbf{\Lambda}_1(\boldsymbol{\mu}_0 - \boldsymbol{\mu}_1) - k + \ln\left(\frac{\det \lambda \mathbf{\Lambda}_1^{-1}}{\det \beta(\mathbf{H}_l + \frac{\beta}{\lambda}\mathbf{\Lambda}_1)^{-1}}\right)) \\
=&\frac{\beta}{2}(\mathrm{tr}((\mathbf{H}_l + \frac{\beta}{\lambda}\mathbf{\Lambda}_1)(\mathbf{H}_l + \frac{\beta}{\lambda}\mathbf{\Lambda}_1)^{-1}) \\
&\frac{1}{\lambda}(\boldsymbol{\mu}_0 - \boldsymbol{\mu}_1)^T \mathbf{\Lambda}_1(\boldsymbol{\mu}_0 - \boldsymbol{\mu}_1) - k + \ln\left(\frac{\det \lambda \mathbf{\Lambda}_1^{-1}}{\det \beta(\mathbf{H}_l + \frac{\beta}{\lambda}\mathbf{\Lambda}_1)^{-1}}\right)) \\
=&\frac{\beta}{2}[+\frac{1}{\lambda}(\boldsymbol{\mu}_0 - \boldsymbol{\mu}_1)^T \mathbf{\Lambda}_1(\boldsymbol{\mu}_0 - \boldsymbol{\mu}_1) + \ln\left(\frac{\det \lambda \mathbf{\Lambda}_1^{-1}}{\det \beta(\mathbf{H}_l + \frac{\beta}{\lambda}\mathbf{\Lambda}_1)^{-1}}\right)]
\end{aligned}
\tag{18}
$$

The above matrix equation 18 is difficult to deal with directly. We will therefore use the common diagonal approximation of the Hessian which is more amenable to manipulation. Substituting $\mathbf{\Lambda}_1 = \mathrm{diag}(\Lambda_{11}, \Lambda_{21}, ..., \Lambda_{k1})$ and $\mathbf{H}_l = \mathrm{diag}(h_{1l}, h_{2l}, ..., h_{kl})$ in the above expression we get

$$
C_\beta(\mathcal{D}; P, Q)|_{\mathbf{\Sigma}_0 = \mathbf{\Sigma}_0^*} = \frac{\beta}{2}(\frac{1}{\lambda}\sum_i \Lambda_{i1}(\mu_{i0} - \mu_{i1})^2 - \sum_i \ln(\frac{\Lambda_{i1}}{\lambda}) + \sum_i \ln(\frac{h_{il} + \frac{\beta}{\lambda}\Lambda_{i1}}{\beta}))
\tag{19}
$$

The above expression is easy to optimize. We see that the sole stationary point exists at

$$
\Lambda_{i1}^* = \frac{\lambda}{2\beta}[\sqrt{h_{il}^2 + \frac{4\beta h_{il}}{(\mu_{i0} - \mu_{i1})^2}} - h_{il}].
\tag{20}
$$

We now turn to the original objective and calculate it's second derivatives. For our diagonal approximation the original objective turns into a sum of separable functions. We will analyze the behavior of one of them for simplicity. The result applies to all other functions in the sum.

$$
\begin{aligned}
C_\beta(\mathcal{D}; P, Q) =& \sum_i \frac{h_{il}}{2}\nu_{i0} + \sum_i \frac{\beta}{2\lambda}\frac{\nu_{i0}}{\nu_{i1}} - \sum_i \frac{\beta}{2} + \sum_i \frac{\beta(\mu_{i0} - \mu_{i1})^2}{2\lambda}\frac{1}{\nu_{i1}} \\
&+ \frac{\beta}{2}[\sum_i \ln(\lambda\nu_{i1}) - \sum_i \ln(\nu_{i0})] \\
=& \sum_i A_i \nu_{i0} + \sum_i B_i \frac{\nu_{i0}}{\nu_{i1}} - \sum_i \frac{\beta}{2} + \sum_i C_i \frac{1}{\nu_{i1}} + D_i[\sum_i \ln(\lambda\nu_{i1}) - \sum_i \ln(\nu_{i0})]
\end{aligned}
\tag{21}
$$

where we have set $A_i = \frac{h_{il}}{2}$, $B_i = \frac{\beta}{2\lambda}$, $C_i = \frac{\beta(\mu_{i0} - \mu_{i1})^2}{2\lambda}$, $D_i = \frac{\beta}{2}$. Denoting $C_{i\beta}(\mathcal{D}; P, Q)$ one function from this sum we calculate

$$\frac{\partial C_{i\beta}(\mathcal{D};P,Q)}{\partial \nu_{i0}} = A_i + \frac{B_i}{\nu_{1i}} - \frac{D_i}{\nu_{i0}}, \quad \frac{\partial C_{i\beta}(\mathcal{D};P,Q)}{\partial \nu_{i1}} = -\frac{B_i \nu_{i0}}{\nu_{i1}^2} - \frac{C_i}{\nu_{i1}^2} + \frac{D_i}{\nu_{i1}} \tag{22}$$

and

$$\frac{\partial C_{i\beta}(\mathcal{D};P,Q)}{\partial^2 \nu_{i0}} = \frac{D_i}{\nu_{i0}^2}, \quad \frac{\partial C_{i\beta}(\mathcal{D};P,Q)}{\partial^2 \nu_{i1}} = 2(B_i \nu_{i0} + C_i)\frac{1}{\nu_{i1}^3} - \frac{D_i}{\nu_{i1}^2} \tag{23}$$

$$\frac{\partial C_{i\beta}(\mathcal{D};P,Q)}{\partial \nu_{i0} \partial \nu_{i1}} = -\frac{B_i}{\nu_{i1}^2}, \quad \frac{\partial C_{i\beta}(\mathcal{D};P,Q)}{\partial \nu_{i1} \partial \nu_{i0}} = -\frac{B_i}{\nu_{i1}^2} \tag{24}$$

We need to check whether the Hessian matrix is PSD so that the stationary point we found is a local minimum and the function is convex. We do that by calculating whether all principal minors of the Hessian are positive.

$$\nabla^2 C_{i\beta}(\nu_{i0}, \nu_{i1}) = \begin{bmatrix} \frac{D_i}{\nu_{i0}^2} & -\frac{B_i}{\nu_{i1}^2} \\ -\frac{B_i}{\nu_{i1}^2} & 2(B_i \nu_{i0} + C_i)\frac{1}{\nu_{i1}^3} - \frac{D_i}{\nu_{i1}^2} \end{bmatrix} \tag{25}$$

We see easily that $\det(\frac{D_i}{\nu_{i0}^2}) > 0$. While

$$\begin{aligned}
\det(\nabla^2 C_{i\beta}(\nu_{i0}, \nu_{i1})) &= \frac{D_i}{\nu_{i0}^2}\left(2(B_i \nu_{i0} + C_i)\frac{1}{\nu_{i1}^3} - \frac{D_i}{\nu_{i1}^2}\right) - \frac{B_i^2}{\nu_{i1}^4} \\
&= \frac{1}{\nu_{i0}^2 \nu_{i1}^4}\left(2C_i D_i \nu_{i1} - (D_i \nu_{i1} - B_i \nu_{i0})^2\right) \\
&= \left(\frac{1}{\nu_{i0}^2 \nu_{i1}^4}\frac{\beta^2}{2}\right)\left(\frac{(\mu_{i0} - \mu_{i1})^2}{\lambda}\nu_{i1} - \frac{1}{2}(\nu_{i1} - \frac{\nu_{i0}}{\lambda})^2\right)
\end{aligned} \tag{26}$$

A first observation is that this determinant is not always positive and the function is not convex everywhere. However we observe that it is not highly non convex either and the non convexity mainly results from the function tending to infinity logarithmically on one of the boundaries. We now check whether the sole stationary point is always a local minimum. We start by substituting $\nu_{i0}^\star = \beta(h_{il} + \frac{\beta}{\lambda}\frac{1}{\nu_{i1}})^{-1}$ in the multiplicand of 26 as the multiplier is positive by definition

$$\begin{aligned}
\det(\nabla^2 C_{i\beta}(\nu_{i0}^\star, \nu_{i1})) &= \frac{1}{\nu_{i0}^{\star 2} \nu_{i1}^4}\frac{\beta^2}{2}\left(\frac{(\mu_{i0} - \mu_{i1})^2}{\lambda}\nu_{i1} - \frac{1}{2}(\nu_{i1} - \frac{\beta}{\lambda}(h_{il} + \frac{\beta}{\lambda}\frac{1}{\nu_{i1}})^{-1})^2\right) \\
&= \frac{1}{\nu_{i0}^{\star 2} \nu_{i1}^4}\frac{\beta^2}{2}\left(\frac{(\mu_{i0} - \mu_{i1})^2}{\lambda}\nu_{i1} - \frac{1}{2}(\nu_{i1} - \frac{\beta}{\lambda}(\frac{\nu_{i1}\lambda}{h_{il}\lambda\nu_{i1} + \beta}))^2\right) \\
&= \frac{1}{\nu_{i0}^{\star 2} \nu_{i1}^4}\frac{\beta^2}{2}\left(\frac{(\mu_{i0} - \mu_{i1})^2}{\lambda}\nu_{i1} - \frac{\nu_{i1}^2}{2}(1 - (\frac{\beta}{h_{il}\lambda\nu_{i1} + \beta}))^2\right) \\
&= \frac{1}{\nu_{i0}^{\star 2} \nu_{i1}^3}\frac{\beta^2}{2}\left(\frac{(\mu_{i0} - \mu_{i1})^2}{\lambda} - \frac{\nu_{i1}}{2}(\frac{h_{il}\lambda\nu_{i1}}{h_{il}\lambda\nu_{i1} + \beta})^2\right) \\
&= \frac{1}{\nu_{i0}^{\star 2} \nu_{i1}^3}\frac{\beta^2}{2}\left(\frac{(\mu_{i0} - \mu_{i1})^2}{\lambda} - \frac{\lambda^2 h_{il}^2 \nu_{i1}^3}{2(h_{il}\lambda\nu_{i1} + \beta)^2}\right) \\
&= \frac{1}{\nu_{i0}^{\star 2} \nu_{i1}^3 2\lambda(h_{il}\lambda\nu_{i1} + \beta)^2}(2(\mu_{i0} - \mu_{i1})^2(h_{il}\lambda\nu_{i1} + \beta)^2 - \lambda^3 h_{il}^2 \nu_{i1}^3) \\
&= \frac{1}{\nu_{i0}^{\star 2} 2\lambda(h_{il}\lambda\Lambda_{i1}^{-1} + \beta)^2}(2\Lambda_{i1}(\mu_{i0} - \mu_{i1})^2(h_{il}\lambda + \Lambda_{i1}\beta)^2 - \lambda^3 h_{il}^2)
\end{aligned} \tag{27}$$

Where we substituted $\nu_{i1} = \Lambda_{i1}^{-1}$ as this will make the calculations easier. We now show a useful identity for $\Lambda_{i1}^\star = \frac{\lambda}{2\beta}[\sqrt{h_{il}^2 + \frac{4\beta h_{il}}{(\mu_{i0} - \mu_{i1})^2}} - h_{il}]$

$$(\Lambda_{i1}^{\star})^2 = \frac{\lambda^2}{4\beta^2}\left(h_{il}^2 + \frac{4\beta h_{il}}{(\mu_{i0}-\mu_{i1})^2} - 2h_{il}\sqrt{h_{il}^2 + \frac{4\beta h_{il}}{(\mu_{i0}-\mu_{i1})^2}} + h_{il}^2\right)$$

$$= \frac{\lambda^2}{4\beta^2}\left(2h_{il}\left(h_{il} - \sqrt{h_{il}^2 + \frac{4\beta h_{il}}{(\mu_{i0}-\mu_{i1})^2}}\right) + \frac{4\beta h_{il}}{(\mu_{i0}-\mu_{i1})^2}\right)$$

$$= \frac{h_{il}\lambda}{\beta}\frac{\lambda}{2\beta}\left(\left(h_{il} - \sqrt{h_{il}^2 + \frac{4\beta h_{il}}{(\mu_{i0}-\mu_{i1})^2}}\right) + \frac{2\beta}{(\mu_{i0}-\mu_{i1})^2}\right)$$

$$= \frac{h_{il}\lambda}{\beta}\left(\frac{\lambda}{(\mu_{i0}-\mu_{i1})^2} - \Lambda_{i1}^{\star}\right)$$

(28)

We substitute $\Lambda_{i1} = \Lambda_{i1}^{\star}$ in 27 and again develop only the multiplicand

$$\det(\nabla^2 C_{i\beta}(\nu_{i0}^{\star},\nu_{i1}^{\star})) = \frac{1}{\nu_{i0}^{\star 2}2\lambda(h_{il}\lambda\Lambda_{i1}^{\star -1}+\beta)^2}(2\Lambda_{i1}^{\star}(\mu_{i0}-\mu_{i1})^2(h_{il}\lambda+\Lambda_{i1}^{\star}\beta)^2 - \lambda^3 h_{il}^2)$$

$$= A_i(2\Lambda_{i1}^{\star}(\mu_{i0}-\mu_{i1})^2(h_{il}\lambda+\Lambda_{i1}^{\star}\beta)^2 - \lambda^3 h_{il}^2)$$

$$= A_i(2\Lambda_{i1}^{\star}(\mu_{i0}-\mu_{i1})^2(h_{il}^2\lambda^2 + 2h_{il}\lambda\Lambda_{i1}^{\star}\beta + (\Lambda_{i1}^{\star})^2\beta^2) - \lambda^3 h_{il}^2)$$

$$= A_i(2\Lambda_{i1}^{\star}(\mu_{i0}-\mu_{i1})^2(h_{il}^2\lambda^2 + 2h_{il}\lambda\Lambda_{i1}^{\star}\beta + \frac{h_{il}\lambda}{\beta}\left(\frac{\lambda}{(\mu_{i0}-\mu_{i1})^2}-\Lambda_{i1}^{\star}\right)\beta^2) - \lambda^3 h_{il}^2)$$

$$= A_i(2\Lambda_{i1}^{\star}(\mu_{i0}-\mu_{i1})^2(h_{il}^2\lambda^2 + h_{il}\lambda\Lambda_{i1}^{\star}\beta + \frac{\beta\lambda^2 h_{il}}{(\mu_{i0}-\mu_{i1})^2}) - \lambda^3 h_{il}^2)$$

$$= A_i(2\Lambda_{i1}^{\star}(\mu_{i0}-\mu_{i1})^2(h_{il}^2\lambda^2 + \frac{\beta\lambda^2 h_{il}}{(\mu_{i0}-\mu_{i1})^2}) + 2(\Lambda_{i1}^{\star})^2(\mu_{i0}-\mu_{i1})^2 h_{il}\lambda\beta - \lambda^3 h_{il}^2)$$

$$= A_i(2\Lambda_{i1}^{\star}(\mu_{i0}-\mu_{i1})^2(h_{il}^2\lambda^2 + \frac{\beta\lambda^2 h_{il}}{(\mu_{i0}-\mu_{i1})^2})$$

$$+ 2\frac{h_{il}\lambda}{\beta}\left(\frac{\lambda}{(\mu_{i0}-\mu_{i1})^2}-\Lambda_{i1}^{\star}\right)(\mu_{i0}-\mu_{i1})^2 h_{il}\lambda\beta - \lambda^3 h_{il}^2)$$

$$= A_i(2\Lambda_{i1}^{\star}(\mu_{i0}-\mu_{i1})^2(h_{il}^2\lambda^2 + \frac{\beta\lambda^2 h_{il}}{(\mu_{i0}-\mu_{i1})^2}) + 2\lambda^3 h_{il}^2 - 2h_{il}^2\lambda^2(\mu_{i0}-\mu_{i1})^2\Lambda_{i1}^{\star} - \lambda^3 h_{il}^2)$$

$$= A_i(2\Lambda_{i1}^{\star}(\mu_{i0}-\mu_{i1})^2(h_{il}^2\lambda^2 + \frac{\beta\lambda^2 h_{il}}{(\mu_{i0}-\mu_{i1})^2}) + \lambda^3 h_{il}^2 - 2h_{il}^2\lambda^2(\mu_{i0}-\mu_{i1})^2\Lambda_{i1}^{\star})$$

$$= A_i(2\Lambda_{i1}^{\star}\beta\lambda^2 h_{il} + \lambda^3 h_{il}^2)$$

$$> 0$$

(29)

where we have set $A_i = \frac{1}{\nu_{i0}^{\star 2}2\lambda(h_{il}\lambda(\Lambda_{i1}^{\star})^{-1}+\beta)^2} > 0$. We have used 28 in lines 4 and 7.

Indeed the stationary point is always a local minimum. What remains is to show that there are no other local minima at the boundaries of the domain. From 21 we see that we only need to evaluate expressions of the form $f(\nu_{i0}) = \nu_{i0} - \ln(\nu_{i0})$ and $g(\nu_{i1}) = \frac{1}{\nu_{i0}} + \ln(\nu_{i0})$. By application of L'Hôpital's rule it's easy to show that

$$\lim_{\substack{\nu_{i0}\to 0 \\ \nu_{i1}=ct}} C_{i\beta}(\nu_{i0},\nu_{i1}) = \lim_{\substack{\nu_{i0}\to+\infty \\ \nu_{i1}=ct}} C_{i\beta}(\nu_{i0},\nu_{i1})$$

$$= \lim_{\substack{\nu_{i0}=ct \\ \nu_{i1}\to 0}} C_{i\beta}(\nu_{i0},\nu_{i1}) = \lim_{\substack{\nu_{i0}=ct \\ \nu_{i1}\to+\infty}} C_{i\beta}(\nu_{i0},\nu_{i1}) = +\infty$$

(30)

this concludes the proof.

$\square$

**Lemma 5.1.** Assuming the following empirical loss $\hat{L}(\boldsymbol{\theta}) = ||f_{\boldsymbol{\theta}}(\mathbf{X}) - \mathbf{Y}||_F$ with $\mathbf{X} = [\boldsymbol{x}_0, ..., \boldsymbol{x}_N]$ and $\mathbf{Y} = [\boldsymbol{y}_0, ..., \boldsymbol{y}_N]$ the following is an upper bound on the IB Lagrangian given that we are at a local minimum:

$$C_\beta(\mathcal{D}; P, Q) \lesssim \sum_l \sqrt{\sum_j c_{lj} \mathop{\mathbb{E}}_{\boldsymbol{\eta} \sim Q'_{lj}} [\frac{1}{2} \boldsymbol{\eta}^T \mathbf{H}_{lj} \boldsymbol{\eta}]} + \beta \sum_{l,j} \text{KL}(Q_{lj} || P_{lj}), \tag{31}$$

where $l$ denotes different layers, $j$ denotes the different neurons at each layer (we assume the same number for simplicity), $\mathbf{H}_{lj}$ denotes the local Hessian, and $Q'_{lj}$ is a centered version of $Q_{lj}$. The local Hessian can be computed efficiently as $\mathbf{H}_{lj} = \frac{1}{N} \sum_{i=1}^N \boldsymbol{z}_{l-1}^i \boldsymbol{z}_{l-1}^{i}{}^T$ and $\boldsymbol{z}_{l-1}^i$ is the latent representation input to layer $l$ for signal $i$.

*Proof.* We start by defining a layerwise empirical error $\hat{E}_l(\boldsymbol{\theta}_l) := \frac{1}{N} \sum_{i=1}^N ||\mathbf{W}_l \boldsymbol{z}_{l-1}^i - \boldsymbol{z}_l^i||_2^2$. One can then easily show that $\hat{L}(\boldsymbol{\theta}) \leq \sum_{k=1}^{L-1} \sqrt{\hat{E}_l(\boldsymbol{\theta}_l)} \prod_{k=l+1}^L ||\hat{\boldsymbol{\theta}}_k||_F + \sqrt{\hat{E}_L(\boldsymbol{\theta}_L)}$ Dong et al. (2017) substituting this in the IB Lagrangian we get

$$
\begin{aligned}
C_\beta(\mathcal{D}; P, Q) &= \mathop{\mathbb{E}}_{\boldsymbol{\theta} \sim Q}[\hat{L}(\boldsymbol{\theta})] + \beta \text{KL}(Q||P) \\
&\leq \mathop{\mathbb{E}}_{\boldsymbol{\theta} \sim Q}[\sum_{l=1}^{L-1} \sqrt{\hat{E}_l(\boldsymbol{\theta}_l)} \prod_{k=l+1}^L ||\hat{\boldsymbol{\theta}}_k||_F + \sqrt{\hat{E}_L(\boldsymbol{\theta}_L)}] + \beta \text{KL}(Q||P) \\
&\leq \sum_{l=1}^{L-1} \sqrt{\mathop{\mathbb{E}}_{\boldsymbol{\theta} \sim Q}[\hat{E}_l(\boldsymbol{\theta}_l)]} \prod_{l=k+1}^L \mathop{\mathbb{E}}_{\boldsymbol{\theta} \sim Q}[||\hat{\boldsymbol{\theta}}_l||_F] + \sqrt{\mathop{\mathbb{E}}_{\boldsymbol{\theta} \sim Q}[\hat{E}_L(\boldsymbol{\theta}_L)]} + \beta \text{KL}(Q||P) \\
&\leq \sum_{l=1}^L c_l \sqrt{\mathop{\mathbb{E}}_{\boldsymbol{\theta} \sim Q}[\hat{E}_l(\boldsymbol{\theta}_l)]} + \beta \text{KL}(Q||P) \\
&\leq \sum_{l=1}^L c_l \sqrt{\mathop{\mathbb{E}}_{\boldsymbol{\eta} \sim Q'}[\left(\frac{\partial \hat{E}_l(\boldsymbol{\theta}_l)}{\partial \boldsymbol{\theta}_l}\right)^T \boldsymbol{\eta} + \frac{1}{2} \boldsymbol{\eta}^T \mathbf{H}_l \boldsymbol{\eta} + O(||\boldsymbol{\eta}||^3)]} + \beta \text{KL}(Q||P) \\
&\approx \sum_{l=1}^L c_l \sqrt{\mathop{\mathbb{E}}_{\boldsymbol{\eta} \sim Q'}[\frac{1}{2} \boldsymbol{\eta}^T \mathbf{H}_l \boldsymbol{\eta}]} + \beta \text{KL}(Q||P)
\end{aligned}
\tag{32}
$$

were in line 3 we use the linearity of expectation, Hölder's inequality due to the non-negativity of the random variables, and Jensen's inequality for the concave square root. In line 4 we hide the Frobenius terms into constants $c_l$. Each error term $\hat{E}_l(\boldsymbol{\theta}_l)$ is only multiplied with Frobenius norm terms $||\hat{\boldsymbol{\theta}}_l||_F$ from the deeper layers. Therefore one can start optimizing from the final layer and proceed to the first while considering $c_l$ as constant. In practice we will just consider all $c_l$ as unknown scaling factors. In line 5 we expand each $\hat{E}_l(\boldsymbol{\theta}_l)$ term using a Taylor expansion, and subsequently ignore the first term as the DNN is assumed to be well trained and the first derivative will be zero, while terms with order higher than 2 are unimportant. We also use $Q'$ to denote the centered version of distribution $Q$.

Taking the first and second derivatives of the layerwise error with respect to $\mathbf{W}_l$ we get

$$\frac{\partial E_l(\boldsymbol{\theta})}{\partial \mathbf{W}_l} = \frac{1}{N} \sum_{i=1}^N \frac{\partial}{\partial \mathbf{W}_l} ||\mathbf{W}_l \boldsymbol{z}_{l-1}^i - \boldsymbol{z}_l^i||_2^2 = \frac{1}{N} \sum_{i=1}^N (\mathbf{W}_l \boldsymbol{z}_{l-1}^i - \boldsymbol{z}_l^i) 2 \boldsymbol{z}_{l-1}^{i}{}^T \tag{33}$$

$$\frac{\partial^2 E_l(\boldsymbol{\theta})}{\partial \mathbf{W}_l \partial \mathbf{W}_l^{(j,:)}} = \frac{1}{N} \sum_{i=1}^N \boldsymbol{z}_{l-1}^i \boldsymbol{z}_{l-1}^{i}{}^T \tag{34}$$

Where the second derivative is with respect to any row $\mathbf{W}_l^{(j,:)}$ of the weight matrix $\mathbf{W}_l$. We see that the full Hessian matrix $\mathbf{H}_l = \frac{\partial^2 E_l(\boldsymbol{\theta})}{\partial^2 \mathbf{W}_l}$ then has a block diagonal structure where each block is equal to $\mathbf{H}_{lj} = \frac{1}{N} \sum_{i=1}^N \boldsymbol{z}_{l-1}^i \boldsymbol{z}_{l-1}^{i}{}^T$. Each row $\mathbf{W}_l^{(j,:)}$ corresponds to a neuron of the layer and for

an appropriate choice of prior and posterior with block diagonal covariances it is easy to see that the final form of expression 32 factorizes as

$$C_\beta(\mathcal{D}; P, Q) \lesssim \sum_l \sqrt{\sum_j c_{lj} \mathop{\mathbb{E}}_{\boldsymbol{\eta} \sim Q'_{lj}} [\frac{1}{2} \boldsymbol{\eta}^T \mathbf{H}_{lj} \boldsymbol{\eta}]} + \beta \sum_{l,j} \mathrm{KL}(Q_{lj} || P_{lj}) \tag{35}$$

this completes the proof. $\square$

