# OpenReview forum: "On PAC-Bayes Bounds for Deep Neural Networks using the Loss Curvature"
_ICLR.cc/2020/Conference — Reject_

### Official Review · AnonReviewer1 · 2019-10-20
**Official Blind Review #1**

**Rating:** 1

**Review:**

The authors  replace the empirical risk term in a PAC-Bayes bound by its second-order Taylor series approximation, obtaining an approximate (?)  PAC-Bayes bound that depends on the Hessian. Note that the bound is likely overoptimistic unless the minimum is quadratic. They purpose to study SGD by centering the posterior at the weights learned by SGD. The posterior variance that minimizes this approximate PAC-Bayes bound can then be found analytically. They also solve for the optimal prior variance (assuming diagonal Gaussian priors/posteriors), producing a hypothetical "best possible bound" (at least under the particular choices of priors/posteriors, and under this approximation of the empirical risk term). The authors evaluate their approximate bound and "best bound possible" empirically on MNIST and CIFAR. This requires  computing approximations of the Hessian for small fully connected neural networks trained on MNIST and CIFAR10. There are some nice visualizations (indeed, these may be one of the most interesting contributions.)

The direction taken by the authors is potentially interesting. However, there are a few issues that would have to be addressed carefully for me to recommend acceptance. First, the comparison to (some very) related work is insufficient, and so the actual novelty is misrepresented (see detailed comments below). Further, the paper is full of questionable vague claims and miscites/attributes other work. At the moment, I think the paper is below the acceptance threshold: the authors need to read and understand (!) related work, and expand their theoretical and/or empirical results to produce a contribution of sufficient novelty/impact.

DETAILED FEEDBACK.

I believe the authors missed some related work by Tsuzuki, Sato and Sugiyama (2019), where a PAC-Bayes bound was derived in terms of the Hessian, via a second-order approximation. How are the results presented in this submission relate to Tsuzuki et al approach?

When the posterior, Q, is a Gaussian (or any other symmetric distribution), \eta^T H \eta is the so-called Skilling-Hutchinson trace estimator. Thus E(\eta^T H \eta) is the Trace(H) scaled by the variance of \eta. The authors seem to have completely missed this connection, which simplifies the final expression considerably.

Why is the assumption that the higher order terms are negligible reasonable? Citation or experiments required.

Regarding the off-diagonal Hessian approximation: how does the proposed layer-wise approximation relate to k-FAC (Martens and Grosse 2015)?

IB Lagrangian: I am not sure why the authors state the result in Thm 4.2 as a lower bound on the IB Lagrangian. What’s the significance of having a lower bound on IB Lagrangian?

Other comments:

Introduction: “At the same time neither the non-convex optimization problem solved in .. nor the compression schemes employed in … are guaranteed to converge to a global minimum.”. This is true but it is really not clear what the point being made is. Essentially, so what? Note that PAC-Bayes bounds hold for all posteriors, even ones not centered at the global minimum (of any objective). The claims made in the rest of the paragraph are also questionable and their purposes are equally unclear. I would be grateful if the authors could clarify.

First sentence of Section 3.1: “As the analytical solution for the KL term in 1 obviously underestimates the noise robustness of the deep neural network around the minimum...”. I have no idea what is being claimed here. The statement needs to be made much less vague.  Please explain.

Section 4: “..while we will be minimizing an upper bound on our objective we will be referring with a slight abuse of terminology to our results as a lower bound.”. I would appreciate if the authors could clarify what they mean here.

Section 4.1 beginning: “We make the following model  assumptions...”. Choosing a Gaussian prior and posterior is not an assumption. It's simply a choice. The PAC-Bayes bound is valid for any choices of Gibbs classifiers. On the other hand, it is an assumption that such distributions will yield "tight" bounds, related to the work of Alquier et al.

Section 4.1 “In practice we perform a grid search over the parameters..”. The authors should mention that such a search should be accounted for via a union bound (or otherwise). The "cost" of such a union bound should be discussed.

The empirical risk of Q is computed using 5 MCMC samples. This seems like a very low number, as it would not even give you one decimal point of accuracy with reasonable confidence! The authors should either use more samples, or account for the error in the upper bound using a confidence interval derived from a Chernoff bound.

Section 4.2: “The concept of a valid prior has been formalized under the differential privacy setting...”. I am not sure what the authors mean by that.

Section 5: “There is ambiguity about the size of the Hessians that can be computed exactly.” What kind of ambiguity?

Same paragraph in Section 5 discusses why there are few articles on Hessian computation. The authors claim that “the main problem seems to be that the relevant computations are not well supported...”. This is followed by another comment that is supposed to contrast the previous claim, saying that storing the Hessian is infeasible due to memory requirements. I am not sure how this claim about memory requirements shows a contrast with the claim on computation not being supported.

First sentence in Section 5.1: I believe this is only true under some conditions.

Section 5.1: The authors should explain why they add a damping term, alpha, to the Hessian, and how alpha affects the results.

***
Additional citation issues:

The connections between variational inference, PAC-Bayes and IB Lagrangian have been pointed out in previous work (e.g. Germain, Bach, Lacoste, Lacoste-Julien (2016); Achille and Soatto 2017).

In the introduction, the authors say “...have been motivated simply by empirical correlations with generalization error; an argument which has been criticized …” (followed by a few citations). Note, that this was first criticized in Dziugaite and Roy (2017).

“Both objectives in … are however difficult to optimize for anything but small scale experiments.”. It seems peculiar to highlight this, since the approach that the authors are presenting is actually more computationally demanding.

Citations for MNIST and CIFAR10 are missing.

***
Minor:
Theorem 3.1 “For any data distribution over..”, I think it was meant to be \mathcal{X} \times  (and not \in )
Theorem 4.2: “For our choice of Gaussian prior and posterior, the following is a lower bound on the IB-Lagrangian under any Gaussian prior covariance”. I assume only the mean of the Gaussian prior is fixed.


Citations are misplaced (breaking the sentences, unclear when the paper of the authors are cited).
There are many (!) missing commas, which makes some sentences hard to follow.

***
Positive feedback: I thought the visualizations in Figure 2 and 3 were quite nice.


**Experience Assessment:**

I have published in this field for several years.

**Review Assessment: Checking Correctness Of Derivations And Theory:**

I carefully checked the derivations and theory.

**Review Assessment: Checking Correctness Of Experiments:**

I assessed the sensibility of the experiments.

**Review Assessment: Thoroughness In Paper Reading:**

I read the paper thoroughly.

---

> ### Author Response · Authors · 2019-11-06
> **Reply to Reviewer 1 Part 3**
>
>
> The authors are not aware of literature where a researcher has tried to evaluate when computation of the Hessian can be done exactly, and the authors have not conducted experiments on the subject, which is also outside of the scope of the current work. In most works that we are aware of, exact computation of the Hessian is assumed to be infeasible and approximations are used instead. Recently [11] has claimed that computation of the Hessian is possible in principle but is simply not supported by current autodiff libraries. In short the authors are aware of various claims unsupported by evidence, have not conducted experiments on the subject and therefore cannot comment on it (hence the term "ambiguity"). The next sentence is meant to be complementary to the previous statements in that the authors can instead comment on how much memory an uncompressed Hessian matrix will take up in RAM. An uncompressed Hessian of moderate size, should require approximately 20GB to store, requiring almost certainly some sort of compression or clever manipulation. This is meant to highlight that dealing with the full Hessian for moderate network sizes should be challenging.
>
> The sentence in 5.1 is lifted directly from [11] which has been recently accepted in NIPS 2019. The authors are willing to double check, whether the sentence needs to include specific conditions.
>
> We do not add a dumping term. From equation 5 in section 4.1 calculating the optimal posterior requires inverting a matrix (H+\beta/\lambda \Sigma_1^{-1}). We note that if we choose a zero mean gaussian prior with diagonal and constant covariance this corresponds to inverting a matrix of the form (H+\alpha I) where \alpha  = \beta/\lambda.
>
> ***
> Additional citation issues:
>
> We agree to include the work of (e.g. Germain, Bach, Lacoste, Lacoste-Julien (2016). We do not claim to be the first to find a connection between PAC-Bayes and VI. In fact [7] exploits this connection and we also cite [12] for exactly this reason. If this is not clear we can explain so in the introduction.
>
> We note that the entire paper [7] is a criticism of empirical correlations, being the first to derive non-vacuous bounds. However the works cited by the authors in this text section attack more fundamental elements of current bounds, specifically mainly uniform convergence, which [7] does not address in detail. It is for this reason that they are mentioned separately, although we had to remove the more detailed discussion due to space constraints.
>
> On "Both objectives in … are however difficult to optimize for anything but small scale experiments.". We have presented our arguments and what we consider as benefits of our approach in detail earlier in this reply.
>
>
> [1]Li, Yingzhen, and Yarin Gal. "Dropout inference in bayesian neural networks with alpha-divergences." Proceedings of the 34th International Conference on Machine Learning-Volume 70. JMLR. org, 2017.
> [2]Gal, Yarin, and Zoubin Ghahramani. "Dropout as a bayesian approximation: Representing model uncertainty in deep learning." international conference on machine learning. 2016.
> [3]Tsuzuku, Yusuke, Issei Sato, and Masashi Sugiyama. "Normalized Flat Minima: Exploring Scale Invariant Definition of Flat Minima for Neural Networks using PAC-Bayesian Analysis." arXiv preprint arXiv:1901.04653 (2019).
> [4]Dinh, Laurent, et al. "Sharp minima can generalize for deep nets." Proceedings of the 34th International Conference on Machine Learning-Volume 70. JMLR. org, 2017.
> [5]Kawaguchi, Kenji, Leslie Pack Kaelbling, and Yoshua Bengio. "Generalization in deep learning." arXiv preprint arXiv:1710.05468 (2017).
> [6]Dong, Xin, Shangyu Chen, and Sinno Pan. "Learning to prune deep neural networks via layer-wise optimal brain surgeon." Advances in Neural Information Processing Systems. 2017.
> [7]Dziugaite, Gintare Karolina, and Daniel M. Roy. "Computing nonvacuous generalization bounds for deep (stochastic) neural networks with many more parameters than training data." arXiv preprint arXiv:1703.11008 (2017).
> [8]Zhou, Wenda, et al. "Non-vacuous generalization bounds at the imagenet scale: a PAC-bayesian compression approach." arXiv preprint arXiv:1804.05862 (2018).
> [9]Wu, Anqi, et al. "Deterministic variational inference for robust bayesian neural networks." (2018).
> [10]Neyshabur, Behnam, Srinadh Bhojanapalli, and Nathan Srebro. "A pac-bayesian approach to spectrally-normalized margin bounds for neural networks." arXiv preprint arXiv:1707.09564 (2017).
> [11]Kunstner, Frederik, Lukas Balles, and Philipp Hennig. "Limitations of the Empirical Fisher Approximation." arXiv preprint arXiv:1905.12558 (2019).
> [12]Achille, Alessandro, and Stefano Soatto. "Emergence of invariance and disentanglement in deep representations." The Journal of Machine Learning Research 19.1 (2018): 1947-1980.

---

> > ### Comment · AnonReviewer1 · 2019-11-12
> > **AnonReviewer1 review update**
> >
> > The authors in their response point out that previous PAC-Bayes/compression based bounds are limited since they are not bounding the error of “a different classifier than the original classifier”.  How do the authors address this in their paper? It would be great if the authors could also be more precise what they mean by “original” and “different”.
> >
> > I also fail to see why “two claim about ( c ) ” that the authors prove are siginificant. (c ) is already an approximation of a quantity. What exactly does it tell us? A similar comment applies to  finding a posterior for an approximation, where “the solution we obtained was the best possible under our approximations”. An optimal posterior for a fixed PAC-Bayes prior is already known. Why not to work with that directly? What does your approximation offer?
> >
> > The authors claim that comparison and further work on off-diagonal Hessian approximations is beyond the scope of this submission and is left as future work. Given that the contributions in the paper are minimal (and significance is questionable), I do not think it's unreasonable to expect further comparisons.
> >
> > Overall, I believe the mathematical investigations presented in the paper lack precision, and the writing lacks clarity. In its current state, the paper is well below the acceptance level.

---

> > > ### Author Response · Authors · 2019-11-13
> > > **AnonReviewer1 Reply**
> > >
> > > After training a deep neural network through SGD, we obtain a set of "original" weights that parametrize a function computed by the DNN. Through PAC-Bayes, one then assesses if a related stochastic DNN classifier will generalize. If one assumes a Gaussian posterior with mean equal to the original weights, then the stochastic classifier whose generalization error one bounds through pac bayes is different from the original, given that it is *stochastic* while the original is deterministic. However, one can argue that they are closely related.
> > >
> > > How are they related? A number of works relate flatness of a minimum to generalization, and to PAC-Bayes [3][4] through empirical correlations. Under this view choosing a posterior that is Gaussian and centered at a minimum obtained by SGD, is just a formal way of quantifying if minima are flat. These works imply that, at least in principle, one should be able to prove generalization solely through the flatness of a given minimum.
> > >
> > > In [1] the authors choose the posterior to be a gaussian distribution and then optimize for both the mean and the covariance of this distribution in order to minimize the PAC-Bayes bound directly. This will result in a different posterior mean. Correspondingly, the deterministic function computed by the DNN and parametrized by this mean could be significantly different from the original set of weights. A similar issue exists with [2], where the authors compress a neural network changing the weights and possibly changing significantly the classifier whose complexity is evaluated.
> > >
> > > Therefore, the works [1][2] show non vacuous bounds but the flat minimum intuition is no longer valid, at least to the eyes of the authors, given that the mean of the posterior changes. By assuming a posterior with a different mean, one is again measuring flatness but in a different point of the parameter space.
> > >
> > > As such, the authors consider it as an open problem to test whether one can find a noise distribution centered on the "original weights", with higher variance along directions of the loss that are flat, and that results in non-vacuous bounds. We propose to solve this problem by estimating the Hessian at the "original" minimum and scaling the variance of a Gaussian posterior based on the curvature information provided by the Hessian, adding noise with higher variance to flat directions. One can also, in the case of [1], simply fix the mean of the posterior and optimise the covariance through SGD of the stochastic PAC-Bayes objective, implicitly measuring the flatness of the minimum. However, we believe that our approach has benefits, which relate to the significance of "claim (ii) about (c)".
> > >
> > > Let's say that one estimates the optimal covariance (and corresponding flatness) either through SGD on the stochastic PAC-Bayes objective as in [1] or through the closed form solution that we are proposing. Then how should one interpret the results if the bound is loose or vacuous? PAC-Bayes gives the possibility of choosing a better prior. It might be that we if we find an informative prior, through for example a separate training set, we might get a tighter or non-vauous bound. We think that it's useful to be able to find a closed form solution with respect to the prior covariance and see as a sanity check whether we can get an improvement *in principle* (we focus on the prior covariance given that, setting the prior mean equal to the random DNN initialization is already a very good choice). We think that it is an interesting result that in Cifar experiments we cannot turn a vacuous bound to non-vacuous even by "cheating".
> > >
> > > Motivated by the above we test as a baseline a Gaussian posterior centered on the original weights with diagonal and constant covariance and a prior centered at the random initialization with the same covariance as the posterior. While the same simple prior centered at zero results in vacuous bounds, this simple prior centered at the initialization results in bounds matching our lowerbound and non-vacuous for mnist. While the emphasis in [1] is in *computing* a non-vacuous bound this experiment reveals that most of the gains compared to previous bounds are from a well chosen prior mean and not optimization of the stochastic objective as is implied in [1]. This is in line with work in VI and description length of DNNs [5]. VI succeeds as an encoding scheme for simple Mnist experimetns but fails for more complex Cifar experiments.
> > >
> > > Of course, as the reviewers mentioned, we make some approximations. We can only argue through experiments and citing prior work whether these results translate from the second order approximation of the IB lagrangian back to the PAC-Bayes objective. We presented some arguments regarding this in previous replies.

---

> > > > ### Author Response · Authors · 2019-11-13
> > > > **references**
> > > >
> > > > Regarding the last two points, we respect the opinion of the reviewer and are also planning more experiments which unfortunately cannot be completed adequately as part of this review.
> > > >
> > > > [1] Dziugaite, Gintare Karolina, and Daniel M. Roy. "Computing nonvacuous generalization bounds for deep (stochastic) neural networks with many more parameters than training data." arXiv preprint arXiv:1703.11008 (2017).
> > > >
> > > > [2] Zhou, Wenda, et al. "Non-vacuous generalization bounds at the imagenet scale: a PAC-bayesian compression approach." arXiv preprint arXiv:1804.05862 (2018).
> > > >
> > > > [3] Keskar, Nitish Shirish, et al. "On large-batch training for deep learning: Generalization gap and sharp minima." arXiv preprint arXiv:1609.04836 (2016).
> > > >
> > > > [4] Neyshabur, Behnam, et al. "Exploring generalization in deep learning." Advances in Neural Information Processing Systems. 2017.
> > > >
> > > > [5] Blier, Léonard, and Yann Ollivier. "The description length of deep learning models." Advances in Neural Information Processing Systems. 2018.

---

> ### Author Response · Authors · 2019-11-06
> **Reply to Reviewer 1 Part 2**
>
> As noted by the reviewer optimising a non-convex objective as in [7] and compressing a DNN heuristically as in [8] always results in valid bounds. We note two main weaknesses of such techniques.
>
> 	i) The bounds in both cases hold for a *different* classifier than the original. While this is fine as a research direction, we believe that it is equally important to investigate the complexity of the original classifier. Why is for example an uncompressed network still able to generalize? Does it mean that the loss landscape is extremely flat in most directions? If so does SGD bias the network to lie on very flat minima and does this phenomenon alone result in good generalization? If yes we should in princible be able to find these flat directions add an appropriate noise distribution and get a non-vacoous bound. These are questions that certainly the current literature has not answered definitively and approaches such as [7][8] avoid.
> 	ii) The methods [7][8] result in bounds that are non-vacuous but loose. What is the source of this looseness? It is difficult to tell. It could be that simply optimizing a stochastic objective so as to get a good posterior in [7] has just not been done using the correct hyperparameters (such as SGD step size). In fact a number of works note how tedious is the task of finding proper hyperparameters such as SGD learning rate for the task of VI [9]. At the very least a contribution of our approach is that it trades off an assumed approximation error of the loss with the benefit of not spending time on manual hyperparameter tuning.
>
> On “As the analytical solution for the KL term in 1 obviously underestimates the noise robustness of the deep neural network around the minimum...”, this refers to the work of [10] which derives an analytical solution to the KL term of the PAC Bayes bound. The resulting bound is vacuous by several orders of magnitude. The analytical solution corresponds to defining a Gaussian posterior distribution over the parameters, with a specific choice of variance. This variance is chosen by assuming that noise added to a layer propagates approximately as a product of the spectral norms of the subsequent layers. This is obviously pessimistic and probably underestimates significantly how much noise can be added to different parameters without changing DNN predictions. Hence the above sentence “As the analytical solution for the KL term in 1 obviously underestimates the noise robustness of the deep neural network around the minimum...”.
>
> On “..while we will be minimizing an upper bound on our objective we will be referring with a slight abuse of terminology to our results as a lower bound.”. We make a number of approximations in our analysis. We substitute the PAC-Bayesian objective with the IB-Lagrangian and then the IB-Lagrangian with a second order Taylor expansion. Our theoretical results are formal only for the *second order Taylor expansion* of the IB-Lagrangian. They might or might not hold for the IB-Lagrangian and the PAC-Bayesian objective. We use the term lower bound for all three objective even though it is formal only for the second order Taylor expansion. We believe that our experiments show that it is meaningful also for the original PAC-Bayes objective.
>
> We agree that "modelling assumptions" should be changed to modelling choices.
>
> The union bound cost is identical to the one from [7][10] and negligible. We agree that it should be discussed.
>
> We understand the concern of the reviewer regarding the number of MC samples. We note the works [1][2] where the samples are of the order of 10^1. We also note that in our own experiments a higher number of samples didn't result in significant differences in accuracy.
>
> Under the PAC-Bayes framework the prior cannot depend on the training set, but can depend on the data distribution. To the best of the authors' understanding the training set can be used under the differential privacy setting to derive an imformative prior while ensuring that the prior remains distribution and not training set dependent. The phrase "The concept of a valid prior has been formalized under the differential privacy setting." can indeed be rewritten in a more accurate way.

---

> ### Author Response · Authors · 2019-11-06
> **Reply to Reviewer 1 Part 1**
>
> Thank you for your *very* detailed review.
>
> Looking at the work of Tsuzuki, Sato and Sugiyama (2019)[3] there are some superficial similarities, specifically the authors expand the loss using the Hessian, and apply PAC-Bayes. However we note a number of important differences:
> 	i) The Hessian is assumed to be diagonal, we make no such assumption, at least in Lemma 4.1. .
> 	ii) The authors make a number of choices which we consider fundamentally flawed, and suboptimal. In page 7 between equations 6 and 7 the authors optimize the KL term independently of the approximated loss. Specifically they set the prior variance equal to the posterior variance. In equation 10 they have already assumed arbitrarily that the two variances are the same then reoptimize with respect to both the KL term and the Taylor expansion. Notice also a constant that continues throughout the calculations and is finally omitted in equation 13, this constant is almost surely making the bound vacuous (even after removal the bound remains vacuous as evidenced by the experiments). From equation 9 it is also obvious that implicitly the authors assume the same noise variance in all parameters of each layer. By contrast we assume much more rich noise which can be different for every parameter (it is clear how this is beneficial, clearly not all parameters in a layer have the same relevence). We also derive the true optimal prior, given the taylor expansion.
> 	iii) Finally we note that in our opinion the research direction of the paper [3] will not be fruitful. In [4] the authors attack flatness based measures of complexity by reparametrizing DNNs to have the same GE and arbitrary sharpness at the minimum. Consequently papers such as [3] seek complexity measures that are invariant to these reparametrizations. Following [5] we consider the entire debate to be flawed. Flatness might very well be a sufficient but not necessary condition for good generalization. To the extent that all solutions reached naturally by SGD are flat it suffices to utilize this flatness to prove generalization. Whether the technique fails in contrived counterexamples is irrelevant.
>
>
> We believe the reviewer is refering to the following identity E[\eta^T H \eta] = E[tr(H \eta \eta^T)] = tr(H\Sigma_0) then tr(H\Sigma_0) = \sigma_0tr(H) only if \Sigma_0 is assumed to be diagonal and with constant variance. We do not consider such a restrictive case. We furthermore use this identity (without assuming the covariance to be diagonal) in Appendix B, Lemma 4.1., equations 12. We would appreciate if the reviewer could point out if we can use this identity somewhere else.
>
> We make a number of approximations and do not claim that they are tight. However at least for the case of the second order approximation we note that it has seen extensive use in the DNN compression literature.  Examples of works by well known authors and in prestigious venues include:
>
> i)Dong, Xin, Shangyu Chen, and Sinno Pan. "Learning to prune deep neural networks via layer-wise optimal brain surgeon." Advances in Neural Information Processing Systems. 2017.
> ii)Wang, Chaoqi, et al. "EigenDamage: Structured Pruning in the Kronecker-Factored Eigenbasis." International Conference on Machine Learning. 2019.
> iii)Peng, Hanyu, et al. "Collaborative Channel Pruning for Deep Networks." International Conference on Machine Learning. 2019.
> iv) LeCun, Yann, John S. Denker, and Sara A. Solla. "Optimal brain damage." Advances in neural information processing systems. 1990.
> v) Hassibi, Babak, and David G. Stork. "Second order derivatives for network pruning: Optimal brain surgeon." Advances in neural information processing systems. 1993.
>
> producing state of the art results in a number of cases. Correspondingly the approximation while almost certainly not tight has proven quite useful and meaningful.
>
> We have not conducted and are not aware at this point of a detailed comparison of k-FAC and the layerwise approximation we used which is based on [6]. We consider expanding to more rich approximations of the Hessian as an interesting future research direction.

---

### Official Review · AnonReviewer2 · 2019-10-22
**Official Blind Review #2**

**Rating:** 3

**Review:**

This paper propose a second-order approximation to the empirical loss in the PAC-Bayes bound of random neural networks. Though the idea is quite straightforward, the paper does a good job in discussing related works and motivating improvements.

Two points made about the previous works on PAC-Bayesian bounds for generalization of neural networks (especially Dziugaite & Roy, 2017) are:
* Despite non-vacuous, these results are obtained on "significantly simplified" datasets and remain "significantly loose"
* The mean of q after optimizing the PAC-Bayes bound through variational inference is far different from the weights obtained in the original classifier.

These points are valid. But it's unclear to me that the proposed method fixes any of them. My concerns are summarized below:
* The inequalities are rather arbitrary and not convincing to me. BY Taylor expansion one actually get a lower bound of the right hand side, However the authors write it as first including the higher-order terms, which results in an upper bound, then throwing the higher-order term and arguing the final equation as an approximate upper bound. I believe this can be incorrect when the higher-order terms plays an nonnegligible role.
* The theorems are easy algebras and better not presented as theorems.
* The proposed diagonal and layer-wise approximation to hessian are very rough estimate of the original Hessian and it is not surprising that it doesn't give meaningful approximation of the original bound.
* There is no explicit comparison with previous methods using the same dataset and architecture. It would be much more convincing if the authors include the results of previous works using the same style of figures as Figure 2/3.

Minor:
* I understand using the invalid bound (optimizing prior) as a sanity check. But the presentation in the paper could better be improved by explaining why doing this.
* Do the plots in Figure 2 correspond to the invalid or valid bound?
* Many papers are complaining that Hessian computation is difficult in autodff libs without noticing this is a fundamental limitation of these reverse-mode autodiff libraries and no easy fix exists.
* I believe MCMC is not used and the authors are refering to MC (page 7, first paragraph).


**Experience Assessment:**

I have read many papers in this area.

**Review Assessment: Checking Correctness Of Derivations And Theory:**

I carefully checked the derivations and theory.

**Review Assessment: Checking Correctness Of Experiments:**

I assessed the sensibility of the experiments.

**Review Assessment: Thoroughness In Paper Reading:**

I read the paper at least twice and used my best judgement in assessing the paper.

---

> ### Author Response · Authors · 2019-11-06
> **Reply to Reviewer 2**
>
> Thank you for your detailed review.
>
> Major:
>
> In the paper we make a number of approximations which we do not claim are tight. We substitute the PAC-Bayesian objective with the IB-Lagrangian, we then approximate the IB-Lagrangian using a second order Taylor expansion of the loss. We then prove some formal results for the second order taylor expansion. These might or might not translate to the original PAC-Bayes objective. We believe our experiments show that to some extent the results are meaningful.
>
> Furthermore there is ample evidence in the literature that although a second order Taylor expansion of the loss around a minimum is loose it can be quite informative. In particular there has been a long line of research in the literature of DNN compression where a second order Taylor expansion of the loss has produced state of the art results in parameter number reduction. We refer to the following which include a number of well known researchers and conferences:
>
> i)Dong, Xin, Shangyu Chen, and Sinno Pan. "Learning to prune deep neural networks via layer-wise optimal brain surgeon." Advances in Neural Information Processing Systems. 2017.
> ii)Wang, Chaoqi, et al. "EigenDamage: Structured Pruning in the Kronecker-Factored Eigenbasis." International Conference on Machine Learning. 2019.
> iii)Peng, Hanyu, et al. "Collaborative Channel Pruning for Deep Networks." International Conference on Machine Learning. 2019.
> iv) LeCun, Yann, John S. Denker, and Sara A. Solla. "Optimal brain damage." Advances in neural information processing systems. 1990.
> v) Hassibi, Babak, and David G. Stork. "Second order derivatives for network pruning: Optimal brain surgeon." Advances in neural information processing systems. 1993.
>
> . Correspondingly the approximation while almost certainly not tight has proven quite useful and meaningful.
>
> Concerning substituting the PAC-Bayes objective for the IB-Lagrangian we note that in [1] page 8, section 6, the authors mention that they have used the PAC Bayes bound and the IB-Lagrangian interchangably when optimising and didn't notice a difference in results.
>
> We do not object to changing the theorems to lemmas. We agree that comparison to [1] would be useful (assuming that we fix the posterior mean and only optimise for the variance using non-convex optimisation) however we note that, this requires careful experimentation and hyperparameter tuning which is certainly not trivial.
>
> Minor:
>
> Yes the plots correspond to the invalid prior see section 4.2 page 6 for details. We would be interested in any literature relating to limitations of reverse-mode autodiff libraries. MC is indeed the correct term.
>
> [1]Dziugaite, Gintare Karolina, and Daniel M. Roy. "Computing nonvacuous generalization bounds for deep (stochastic) neural networks with many more parameters than training data." arXiv preprint arXiv:1703.11008 (2017).

---

### Official Review · AnonReviewer3 · 2019-10-28
**Official Blind Review #3**

**Rating:** 1

**Review:**

Summary: The paper provides several approximations of PAC-Bayes generalization bounds for Gaussian prior and posterior distributions, with various restrictions on the covariance matrices.
In particular, the paper:
(1) Assumes that the expectation of the loss can be Taylor expanded around each point in the support, and all but the quadratic (Hessian) term can be ignored.
(2) Proves a lower bound on the PAC-Bayes generalization objective.
(3) Provides an upper bound on the PAC-Bayes objective via a "layerwise" Hessian objective.

Evaluation: I found this paper extremely difficult to follow because it's sloppy in various places -- both in terms of what claims are formal, and what are heuristic approximations -- and in terms of properly defining crucial quantities. I will go in the same numbered order in which I listed the main contributions above:
(1) (Taylor approximation): The equation (4) is an *approximation* -- not a lower or upper bound. Moreover, too little is said about this heuristic: note that the authors actually Taylor expand an *expectation* over \theta -- the trivial thing to require for this Taylor approximation to hold is that it holds for *every* theta which clearly will not be true. It seems the authors want to say the distribution Q concentrates over thetas close to some local optimum \theta^*, and over these thetas the approximation holds. At the very least something needs to be said about how much things need to concentrate and whether this is realistic in real-life settings.
Also, because (4) is an approximation, it's a little disengenuous to call Theorems 4.2 and 5.2 "theorems", and it needs to be mentioned in the statements that they hold under some formalization of the approximation I described above.
(2) The lower bound is written very oddly -- the "prior" for the lower bound is really dependent upon the posterior -- so it is very strange to call it an "invalid" prior. Moreover, I have serious problems evaluating the meaning of this lower bound -- as it uses the Taylor approximation from (1), but then decides to instantiate the prior *depending on the optimum* of this Taylor approximation. As such, *at the very least* -- some small neural net examples should be tried where the normal (un-approximated) KL bound can be evaluated, to check whether this *actually* is a lower bound most of the time.
(3) The upper bound is also written rather sloppily: Q_{lj} is never defined; H_{lj} only depends on l, rather than j -- in fact, I'm fairly sure it should be H_l, and \eta should be sampled from Q_{l} (i.e. a vector with a coordinate for each neuron in layer l) if I understood the proof correctly.


**Experience Assessment:**

I have published in this field for several years.

**Review Assessment: Checking Correctness Of Derivations And Theory:**

I assessed the sensibility of the derivations and theory.

**Review Assessment: Checking Correctness Of Experiments:**

I assessed the sensibility of the experiments.

**Review Assessment: Thoroughness In Paper Reading:**

I read the paper thoroughly.

---

> ### Author Response · Authors · 2019-11-06
> **Reply to Reviewer 3**
>
> Thank you for your detailed review.
>
> 1) The Taylor expansion of the loss is indeed an approximation in general. For the specific case of a DNN solution, assuming that we have reached a local minimum, and that the loss function is locally convex a second order expansion can be seen as an upper bound to the loss. However we do not make this formal, nor do we think that it is easy or useful to state strict conditions on whether the approximation is an upper bound. We point however to a long line of work in the DNN compression literature where a second order Taylor expansion of the loss has led to state of the art results in DNN compression.
>
> i)Dong, Xin, Shangyu Chen, and Sinno Pan. "Learning to prune deep neural networks via layer-wise optimal brain surgeon." Advances in Neural Information Processing Systems. 2017.
> ii)Wang, Chaoqi, et al. "EigenDamage: Structured Pruning in the Kronecker-Factored Eigenbasis." International Conference on Machine Learning. 2019.
> iii)Peng, Hanyu, et al. "Collaborative Channel Pruning for Deep Networks." International Conference on Machine Learning. 2019.
> iv) LeCun, Yann, John S. Denker, and Sara A. Solla. "Optimal brain damage." Advances in neural information processing systems. 1990.
> v) Hassibi, Babak, and David G. Stork. "Second order derivatives for network pruning: Optimal brain surgeon." Advances in neural information processing systems. 1993.
>
> The distribution over parameters in the Gaussian posterior case with diagonal covariance should be highly concentrated around the mean making the approximation meaningful.
>
> 2)We call the prior "invalid" in that through the way we calculate it, it depends on the posterior. This is not allowed in the PAC-Bayes framework, the prior has to be independent of the training set. Note that even though we calculate a invalid posterior based on the second order Taylor expansion of the IB-Lagrangrian, it remains invalid for the PAC-Bayes bound. In section 4.2 we make a detailed discussion, regarding the benefits and limitations of this result. In particular after one has computed an optimal posterior using equation 5, seeing that the result is non-vacuous or loose one may be tempted to search for a better prior, for example through a separate training set. Our result using the invalid prior corresponds to a lower bound for equation 4 (the second order taylor expansion of the IB-Lagrangian). We can trace a corresponding feasible region vs non-vacuous region using this lower bound (Figure 2). Thus if the two regions don't overlap we should not be able to prove generalization even if we chose a better prior in a valid manner. To be precise what we can show is that we cannot minimize the IB-Lagrangian second order approximation further. Ideally we would like these results to translate to the PAC-Bayes theorem directly (we would like the feasible regions to be meaningful for equation 1 even though we calculated them through equation 4), however we cannot prove this formally and have to rely on experiments. In practice we have found the calculated feasible region to be meaningful, the baseline and the diagonal gaussian posteriors fail to cross it. The non-diagonal posterior crosses it slightly.
>
> 3) Q_{lj} = \mathcal{N}(\mu_{0lj},\Sigma_{0lj}) and the dimension number of the multivariate Gaussian is equal to the dimensionality of the input to the layer "l". While H_{l1}=H_{l2}=...=H_{l*} = \frac{1}{N}\sum_i z^i_{l-1}z^i_{l-1}^T it is not true that H_{lj} = H_{l}. H_{lj} is a local Hessian for each neuron therefore has a dimensions equal to layer_input_dims \times layer_input_dims. H_{l} is the local Hessian of the layer and therefore has dimensions equal to (layer_input_dims * layer_output_dims) \times (layer_input_dims * layer_output_dims). H_{l} is block diagonal with blocks  H_{lj}.
>
> Note that the layerwise approximation of section 5.2 is only a heuristic which is independent from the rest of our analysis and is mainly used to motivate solving for the optimal posterior in an layerwise manner, which is in general a quite cheap calculation. In fact one can completely ignore section 5.2 and perform most experiments by computing a diagonal Hessian using equation 9.

---

### Public Comment · ~kento_nozawa1 · 2019-10-23
**Clarify notations**

I'm interested in this practical PAC-Baye bound and its optimization.

I am a little bit confused by some notations while I read this paper;

## Theorem 3.

This \mathcal{X} may be a data distribution to sample training data; not binary.
In addition, it is already defined as the input space on the same page.

## Eq. 2

\mathcal{D} looks undefined.

---

I also report the following typos:

- PAC Baye ->   PAC-Baye (page 2)
- need a space between "sufficiency," and "minimality" on page 2?
- non vacuous -> non-vacuous (page 2)
- Abundant parentheses:  \beta KL((Q|P)).

---

> ### Author Response · Authors · 2019-10-25
> **Clarifications**
>
> Dear Kento Nozawa,
>
> thank you for your thorough reading of our work and for pointing out notations and typos that indeed need to be fixed. We believe that most of the problems mentioned concerning equations 1 and 2 can be corrected by doing the following
>
> 1) Streamlining the notation denoting "training set". \mathcal{D} denotes any training set. Thus all instances of "S" which is also used to denote any training set need to be replaced by  \mathcal{D}.
>
> 2) In theorem 3.1 we believe it suffices to replace "For any distribution over \mathcal{X} \in \{-1,+1\}" with "For any distribution \mathcal{P} on \mathcal{X}\times\mathcal{Y}".
>
> Actually it is already implied that (x,y)\sim\mathcal{P} in section 3 in the paragraph preceding Theorem 3.1, when we define the population loss. Thus this paragraph can also be improved by adding something along the lines of "\mathcal{P} is any distribution on \mathcal{X}\times\mathcal{Y}". Note that since "P" without the calligraphic font is already used to denote the prior of the classifier, another letter to be decided might be more applicable for the aforemented distribution on \mathcal{X}\times\mathcal{Y}.

---

### Author Response · Authors · 2019-11-06
**Reply to all reviewers Part 3.**

------%% Notes on validity of approximations %%-----
In [4] page 8, section 6, the authors mention that they have used the PAC Bayes bound and the IB-Lagrangian interchangably when optimising and didn't notice a difference in results.

At a minimum there has been a long line of research in the literature of DNN compression where a second order Taylor expansion of the loss has produced state of the art results in parameter number reduction. We refer to the following which include a number of well known researchers and conferences:

i)Dong, Xin, Shangyu Chen, and Sinno Pan. "Learning to prune deep neural networks via layer-wise optimal brain surgeon." Advances in Neural Information Processing Systems. 2017.
ii)Wang, Chaoqi, et al. "EigenDamage: Structured Pruning in the Kronecker-Factored Eigenbasis." International Conference on Machine Learning. 2019.
iii)Peng, Hanyu, et al. "Collaborative Channel Pruning for Deep Networks." International Conference on Machine Learning. 2019.
iv) LeCun, Yann, John S. Denker, and Sara A. Solla. "Optimal brain damage." Advances in neural information processing systems. 1990.
v) Hassibi, Babak, and David G. Stork. "Second order derivatives for network pruning: Optimal brain surgeon." Advances in neural information processing systems. 1993.

. Correspondingly the approximation while almost certainly not tight has proven quite useful and meaningful.
----------------------------------------------------

On a more personal level we would request that the reviewers refrain from comments such as "the authors need to read and understand (!) related work" which we don't find conducive to a useful review process. The authors have read extensively the relevant literature and to the extent that any points have been misunderstood are willing to change their perspective as part of a constructive review process.

[1]Neyshabur, Behnam, Srinadh Bhojanapalli, and Nathan Srebro. "A pac-bayesian approach to spectrally-normalized margin bounds for neural networks." arXiv preprint arXiv:1707.09564 (2017).
[2]Bartlett, Peter L., Dylan J. Foster, and Matus J. Telgarsky. "Spectrally-normalized margin bounds for neural networks." Advances in Neural Information Processing Systems. 2017.
[3]Golowich, Noah, Alexander Rakhlin, and Ohad Shamir. "Size-independent sample complexity of neural networks." arXiv preprint arXiv:1712.06541 (2017).
[4]Dziugaite, Gintare Karolina, and Daniel M. Roy. "Computing nonvacuous generalization bounds for deep (stochastic) neural networks with many more parameters than training data." arXiv preprint arXiv:1703.11008 (2017).
[5]Zhou, Wenda, et al. "Non-vacuous generalization bounds at the imagenet scale: a PAC-bayesian compression approach." arXiv preprint arXiv:1804.05862 (2018).
[6]Wu, Anqi, et al. "Deterministic variational inference for robust bayesian neural networks." (2018).
[7]Dziugaite, Gintare Karolina, and Daniel M. Roy. "Data-dependent PAC-Bayes priors via differential privacy." Advances in Neural Information Processing Systems. 2018.
[8]Parrado-Hernández, Emilio, et al. "PAC-Bayes bounds with data dependent priors." Journal of Machine Learning Research 13.Dec (2012): 3507-3531.

---

> ### Author Response · Authors · 2019-11-15
> **Reply to all reviewers Part 4. Paper Revisions.**
>
> The authors have changed significantly the abstract, introduction and contributions sections of the submission, addressing a number of reviewer concerns. In the revised text we highlight that we make a number of approximations to the original PAC-Bayes objective and derive some formal results for this approximation. We have added a number of citations regarding the validity of the second order approximation. Wether our results translate to the original objective PAC-Bayes still needs to be shown through experiments, of which we include 2 architectures and 6 different datasets. The experiment number is in line with the number of experiments presented in previous influential works on PAC-Bayes.
>
> We have stated more clearly the goals of our paper. Previous works mix various modelling choices to obtain non-vacuous but loose bounds, such as non-convex optimization and well motivated priors. While these bounds are valid this obfuscates the contribution of each choice and invalidates flat minima intuition regarding PAC-Bayes. We believe that it is important to see wether the relationship between PAC-Bayes and flat minima can be taken literally without resorting to arguments of empirical correlation. Motivated by our theoretical analysis we see that datasets and architectures can be separated in easy and hard cases, where one can and can't prove generalization even through "cheating" with a data dependent prior. For the easy cases, a simple baseline with a prior centered at the DNN initialization matches our lower bound and is sufficient to prove generalization, contrary to what is implied in previous works.

---

### Author Response · Authors · 2019-11-06
**Reply to all reviewers Part 2**

We are transparent in that we are minimizing an approximation to the PAC-Bayes objective, and we don't wish to claim that the approximation is tight, however that doesn't prevent it from being meaningful. We will discuss this matter at the end of the post in detail. In our approach one has to accept an approximation error, however it (our method) and our results have a number of benefits over previous works.
	i) The optimum for the posterior can be found in closed form and there is no uncertainty about whether the corresponding bound is as tight as possible due to non-convex optimisation that hasn't been done succesfully. Considerable time can be saved by avoiding hyperparameter tuning as in [4]. We are not claiming that our approach will work better than [4], it might not since [4] optimises directly the non-convex objective, however we can be confident that the solution we obtained was the best possible under our approximations, and won't require tuning of hyperparameters.
	ii) Works such as [7] have raised the possibility of computing better priors using the training set, by utilizing the differential privacy framework. In fact one can in theory do the same without differential privacy, using a separate training set[8]. Note that data driven optimization of the prior will probably be non-convex and will require extensive hyperparameter tuning. By contrast we can optimise over the prior and find a closed form "invalid" solution. If the feasible and non-vacuous regions don't overlap as in the case of Cifar in Figure 2, then we can be confident that we cannot prove generalization using our approximations, even if we searched for a better valid prior.
	iii) We analyze the complexity of the *original* classifier. This is impossible in general for [5].

These results (i and ii) hold for (c), one would ideally want these results to hold for (a) as well, at least around the specific weight instantiation corresponding to the DNN who's complexity we are trying to evaluate. We cannot show this formally nor do we think that it is easy to prove. Specifically we have shown a lowerbound on (c) which might or might not correspond empirically to a lower bound on (a). In section 4 we mention "Furthermore while we will be minimizing an upper bound on our objective we will be referring with a slight abuse of terminology to our results as a lower bound." in section 7 we note another problem "Crucially all results depend on high quality estimates of the Hessian which remains an open topic of research for large scale modern deep neural networks.".

We can, however, conduct experiments to see if our theoretical results, have any merit. We conduct experiments with 3 valid posterior choices. The baseline is i.i.d. Gaussian noise on each parameter and *doesn't* rely on any approximation of the objective. The second method is an optimal gaussian with diagonal covariance under our approximation. Both the baseline and method number 2 which correspond to Gaussian posteriors with diagonal covariance, do not violate our lowerbound *empirically*. Method number 3 which corresponds to a Gaussian with *non-diagonal* covariance seems to violate our lowerbound slightly. More rich posteriors possibly depending on richer approximations of the Hessian might violate the lowerbound further, however we consider this as interesting future work.

---

### Author Response · Authors · 2019-11-06
**Reply to all reviewers Part 1**

We would like to thank all reviewers for their *very* detailed reviews. As noted by reviewer 3 our paper contains a number of heuristic approximations, as well as some formal results. In the submission text, we are confident that we have taken care to state our claims modestly and to include disclaimers whenever heuristic approximations were made. However these scattered disclaimers seem to have confused the reviewers, leading them to assume that we are overclaiming our results. With hindsight a clear discussion of approximations and formal results should have been included in the introduction. We include a summary and discussion in this post, which addresses a number of concerns by the reviewers and we will address more specific criticisms individually for each reviewer.

The earliest use of PAC-Bayes in the modern era of *deep* neural networks (post-2012) is to the best of our knowledge the work of Neyshabur et al.[1]. The problem with this bound as well as other analytically derived bounds [2][3] is that it is vacuous by several orders of magnitude.

Concequently at least two works [4][5] have used heuristic compression of DNNs [5] as well as Variational Inference(VI) style optimisation of a stochastic DNN [4] to find non-vacuous bounds. We believe that these works have two main limitations:
	i) The bounds derived by these methods apply to a *different classifier* than the original. Compressing a DNN corresponds to finding a completely different point in the parameter space than the original minimum. At the same time optimising a stochastic DNN as in [4] where the means of the stochastic parameters change, also corresponds in finding a different minimum in the parameter space.
	ii) As correctly mentioned by reviewer 1 it is true that the bounds are valid given any compression solution or optimization solution. However in practice these bounds are not only non-vacuous but also quite loose. An issue then arises given this looseness. How can one be sure that the looseness is the result of the proof technique and not incorrect compression or optimisation of a stochastic objective? Furthermore considerable time can be wasted in hyperparameter tuning both in compressing a DNN adequately but more importantly getting a stochastic objective as in [4] to converge adequately (see [6] for a detailed discussion). Even then one cannot be confident that the bound is as tight as possible.

.Our paper then makes a number of heuristic approximations which correspond roughly to the following hierarchy.

(a)PAC Bayes bound -> (b)IB-Lagrangian -> (c)Taylor expansion of IB-Lagrangian -> (d)Layerwise upperbound and Taylor expansion of IB-lagrangian

We then can prove formally two claims about (c):
	i) (c) is convex with respect to \Sigma_0 and the global minimum can be found in closed form.
	ii) (c) is non-convex with respect to \Sigma_0 and \Sigma_1 jointly however we can find the global minimum with respect to both these variables as long as they are diagonal. We call this result an invalid solution and the corresponding prior an invalid prior. This is because under the PAC-Bayes framework the prior cannot depend on the training data. Otherwise one could chose the prior to be equal to the posterior and the KL term would be easily equal to zero.

---

### Decision · Program_Chairs · 2019-12-19

**Decision:**

Reject

**Comment:**

The paper computes an "approximate" generalization bound based on loss curvature. Several expert reviewers found a long list of issues, including missing related work and a sloppy mix of formal statements and heuristics, without proper accounting of what could be gleaned from some many heuristic steps. Ultimately, the paper needs to be rewritten and re-reviewed.